# HypoBootstrap: A Bootstrapping Framework for Inductive Reasoning

**Si Chen**[†]**, Yifei Li**[†]**, Richong Zhang**[*]
SKLCCSE, Beihang University, Beijing, China
chen.si@buaa.edu.cn, lyf0862@buaa.edu.cn, zhangrc@act.buaa.edu.cn

## Abstract

Inductive reasoning infers general rules from observed evidence, which is one of the most critical intelligence abilities. Previous works have succeeded in formal languages but suffer from onerous and error-prone conversions between a particular formal language and the working language. As large language models (LLMs) have emerged, direct reasoning with various kinds of languages, especially natural languages, without formal language involvement has become feasible. However, existing LLM-based inductive reasoning usually relies on LLM's intrinsic generation ability, which is prone to LLM's hallucination and lacks systematic guidance according to the nature of inductive reasoning. To this end, we propose HypoBootstrap, an integrated framework for inductive reasoning that generates and confirms hypotheses both in a bootstrapping manner. Regarding hypothesis generation, we propose a novel bootstrapping generation strategy, bootstrapping object hypotheses, relational hypotheses, and functional hypotheses successively, which assists LLM in observing the evidence from trivial patterns to non-trivial patterns. Regarding hypothesis confirmation, we utilize Glymour's theory of bootstrap confirmation, a hypothesis confirmation theory from the philosophy of science that can confirm a set of hypotheses. We use its principles to confirm the object hypotheses, relational hypotheses, and functional hypotheses. Empirical studies on four inductive reasoning scenarios of different natures, involving causal induction, concept learning, grammar learning, and abstract reasoning, demonstrate that HypoBootstrap significantly outperforms existing methods.

## 1  Introduction

Inductive reasoning emblems high-level human intelligence [8, 19, 11, 1]. While deductive reasoning is a critically required ability for intelligence, inductive reasoning is more significant, by which means humans infer general rules from observed evidence. These rules are the basis of the soundness of deductive reasoning. Figure 1 illustrates an inductive reasoning task. In contrast to deductive reasoning, where we reason over given rules, inductive reasoning has no *rules* to follow and requires creative thinking to generate various possible hypotheses.

In the past few decades, Inductive Logic Programming (ILP) [2] was the major research subject for inductive reasoning in the AI community. An ILP system can learn symbolic rules from symbolic evidence, typically expressed in first-order language. An example hypothesis in first-order language is depicted in Figure 1. While ILP can efficiently learn interpretable rules in formal language, it mandatorily requires the formalization of evidence, preventing it from inductive reasoning over more common natural-language evidence. Besides, since ILP cannot produce natural-language rules, translating formal-language rules into natural language is another tremendous burden.

---

[†]Equal contribution
[*]Corresponding author

39th Conference on Neural Information Processing Systems (NeurIPS 2025).

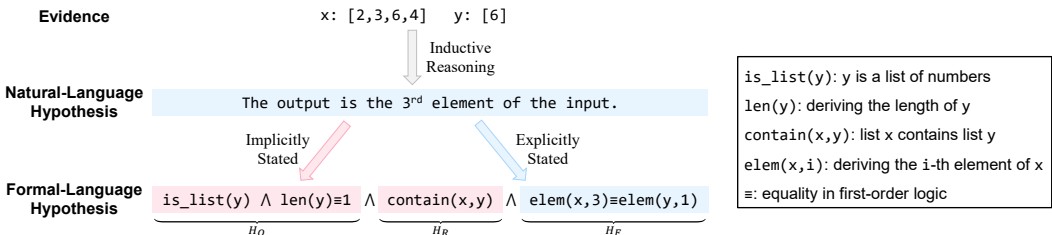

Figure 1: Comparison of natural-language hypothesis with formal-language hypothesis given the same evidence. Natural-language hypotheses are typically imprecise and implicitly imply some statements, which are usually considered trivial and redundant in daily conversation. In contrast, formal-language hypotheses are as precise as possible; even trivial things should be expressed explicitly. HypoBootstrap first generates the trivial hypotheses (i.e., object hypothesis $H_O$ and relational hypothesis $H_R$), which are much easier to observe, and final hypotheses (i.e., functional hypothesis $H_F$) are generated based on these hypotheses.

With the rise of large language models (LLMs; 28), direct reasoning in non-formal languages becomes feasible, especially in natural languages, which avoid the troublesome mutual conversion between non-formal language and formal language. Existing works reveal some degree of ability to propose natural-language hypotheses via LLM's powerful generation ability [20, 25]. However, due to the unavoidable hallucination, LLMs usually unconsciously propose fallacious hypotheses, losing the rigorous nature of formal-language reasoning.

To alleviate LLM's hallucination and provide systematic guidance for inductive reasoning, we propose HypoBootstrap, a novel agent framework to boost LLM's inductive reasoning performance. Hypo-Bootstrap enhances inductive reasoning from two tightly interacting aspects, hypothesis generation (i.e., hypothesis proposing) and hypothesis confirmation, the two indispensable aspects of inductive reasoning. On the one hand, HypoBootstrap enhances hypothesis generation by bootstrapping hypotheses from trivial to non-trivial. The trivial hypotheses, corresponding to the implicitly stated hypotheses of Figure 1 (pink), are easy to observe and help generate more non-trivial hypotheses, corresponding to the explicitly stated hypotheses of Figure 1 (blue). Specifically, inspired by first-order languages, HypoBootstrap successively generates object hypotheses, relational hypotheses, and functional hypotheses, bootstrapping the latter from the former. On the other hand, HypoBootstrap confirms hypotheses based on Glymour's theory of bootstrap confirmation [6], a widely-used confirmation theory from the philosophy of science that is advantageous to confirm a set of hypotheses. HypoBootstrap adopts this confirmation method to confirm the set of object hypotheses, relational hypotheses, and functional hypotheses. Implementing such a confirmation theory originated from first-order languages via LLM agents is not direct; hence, we deliberately design a confirmation procedure and simultaneously integrate it into our bootstrapping generation procedure, which can alleviate the LLM's hallucination during hypothesis generation.

We conduct experiments [1] on four inductive reasoning scenarios with varying nature: causal induction, concept learning, grammar learning, and abstract reasoning. Empirical results demonstrate significant improvement of HypoBootstrap compared with previous works and verify that both the bootstrap generation and bootstrap confirmation are effective. In addition, HypoBootstrap can be further improved via human priors, demonstrating its broader potential in inductive reasoning. The main limitation of HypoBootstrap is the high inference cost. In practice, the trade-off between performance and cost should be considered seriously.

## 2 Related work

### 2.1 Inductive reasoning via language models

Language model-based inductive reasoning enables inferring hypotheses in various languages, especially natural languages, and does not require formalizing evidence. Existing research is divided into

---

[1]Code is available at https://github.com/chensi99/HypoBootstrap.

two lines. One line requires parameter fine-tuning [24], which is costly for LLMs, even impossible for closed-source but powerful models. The fine-tuning may even require annotated rules as training data [22, 26], which is costly, error-prone, and intractable for broader applications. Another line does not rely on fine-tuning, where language models are directly prompted to infer rule hypotheses from evidence, as in Hypothesis Refinement [20]. The hypotheses are confirmed via programs, which are translated from natural-language hypotheses via LLMs. The hypotheses can be refined by iteratively generating with error feedback. Hypothesis Search [25] attempts to inject human priors into prompts, which can further stimulate LLMs' inductive reasoning ability. MoC [13] prompts LLM first to generate relevant concepts for each inductive reasoning task, and then to generate hypotheses according to these concepts, which enriches the diversity of generated hypotheses. However, none of these works explicitly alleviates hallucinations during hypothesis generation. Our work falls into the second line and alleviates LLM's hallucination in inductive reasoning. Further, our proposed framework considers the nature of inductive reasoning and provides systematic guidance for LLMs.

## 2.2 Confirmation theory

Confirmation theory studies the logic that a hypothesis may be confirmed or disconfirmed by given evidence. Various confirmation theories are used in the philosophy of science, such as the Hempelian positive-instance account of confirmation, the hypothetico-deductive method, and the probabilistic method [17]. Glymour's theory of bootstrap confirmation [6] is the one that is, implicitly or explicitly, often used in various significant scientific breakthroughs, such as Newton's universal gravitation, general theory of relativity [6], and quantum theory [15, 16]. We adopt Glymour's bootstrap confirmation since it is designed and appropriate for scenarios where multiple hypotheses should be confirmed together.

# 3 Preliminary

## 3.1 Inductive reasoning

Following previous works [20, 25], we consider a practical induction problem of inferring a mapping rule from an input space $\mathcal{X}$ to an output space $\mathcal{Y}$. All the elements of the spaces are assumed to be represented in some language, and all the inputs and outputs passed to LLMs refer to their language representation. To have coherent notations throughout the paper, we deliberately express this problem in first-order logic. [2] Specifically, for each task, we have observed evidence, a consistent set of binary ground atoms $E = \{f(x^{(1)}) \equiv y^{(1)}, \ldots, f(x^{(n)}) \equiv y^{(n)}\}$, where $x^{(1)}, \ldots, x^{(n)} \in \mathcal{X}$, $y^{(1)}, \ldots, y^{(n)} \in \mathcal{Y}$, $f$ is the mapping function to be inferred, and $\equiv$ denotes the binary predicate *equality*. The goal is to infer a rule $H$ s.t. $\forall x \in \mathcal{X}, y \in \mathcal{Y}$,

$$H \vdash f(x) \equiv y \qquad \text{if } f(x) \text{ is equal to } y,$$
$$H \vdash \neg f(x) \equiv y \qquad \text{if } f(x) \text{ is not equal to } y.$$

For instance, given evidence $\{f([4, 2, 3, 1]) \equiv [1, 2, 3, 4]\}$, where input and output are both lists of numbers, we may infer $H$ as *y is the sorted list of x in ascending order*. The evaluation of a rule is conducted on unobserved evidence, a separate set of binary ground atoms generated from the same mapping $f$. The hold-out evidence ensures that $E$ itself would not be an acceptable rule and a good inductive reasoner should have inferred the underlying mapping behind the observed evidence.

The hypotheses and evidence involved throughout the paper are all supposed to be expressible in a first-order language, though they are never explicitly converted into it; i.e., we conduct induction directly in various working languages that modern LLMs support, such as natural language. This assumption is acceptable since knowledge has been, to some extent, successfully represented by first-order languages in AI research, such as knowledge graph [9] and logic programming [14]. We also give examples of translating our experimental data into first-order language (Appendix D).

## 3.2 Glymour's theory of bootstrap confirmation

While Glymour's original work [6] leaves some space for interpretation, several formalizations exist. We adopt the formalization from [4] to conduct a rigorous discussion.

---

[2]We assume basic knowledge about first-order logic.

**Definition 3.1** (Glymour's Bootstrap Confirmation (simplified version)). Given a set of hypotheses $H = \{H_1, \ldots, H_m\}$, the evidence $E$ bootstrap-confirms $H$ if

1. $H \cup E \nvdash \bot$                                                                  (consistency)

2. for each $i \in \{1, \ldots, m\}$, there is a $H' \subsetneq H$ s.t. $H_i \notin H'$ and $E$ *confirms* $H_i$ with respect to $H'$                                                         (confirmation)

3. there is possible, but not observed, evidence $E'$ s.t. $E'$ *disconfirms* $H$        (non-triviality)

See Appendix A for more details. In addition, Definition 3.1 requires an auxiliary (non-bootstrap) confirmation & disconfirmation method. Glymour's theory tends to be neutral across any auxiliary confirmation & disconfirmation. In practice, hypothetico-deductive (HD) confirmation & disconfirmation is one of the common choices, which we defined in Definition 3.2 & 3.3 following [7]. The exact auxiliary confirmation & disconfirmation HypoBootstrap uses is an implementation feasible by LLM and will be illustrated in Section 4.2.2 & 4.2.3.

**Definition 3.2** (Hypothetico-Deductive Confirmation). Given a hypothesis $H$, the evidence $E$ hypothetico-deductively confirms $H$ with respect to $H'$ if

1. $H \cup H' \nvdash \bot$    (consistency)      2. $H \cup H' \vdash E$    (entailment)      3. $H' \nvdash E$    (necessity)

**Definition 3.3** (Hypothetico-Deductive Disconfirmation). Given a hypothesis $H$, the evidence $E$ hypothetico-deductively disconfirms $H$ if

1. $H \nvdash \bot$    (consistency)      2. $H \vdash \neg E$    (entailment)

## 4 HypoBootstrap

This section presents HypoBootstrap, an inductive reasoning framework that utilizes LLM agents to implement hypothesis bootstrap generation and Glymour's bootstrap confirmation. The pseudo-code is given in Appendix B.1. As depicted in Figure 2, the framework generates the object hypothesis, relational hypothesis, and functional hypothesis in a bootstrapping manner, with confirmations embedded into the generation procedure. If not confirmed, the framework iteratively regenerates the functional hypothesis according to error feedback from the last iteration. See Appendix B.3 for implementation details of the agents in Figure 2.

### 4.1 Bootstrap generation

Inspired by first-order languages that mainly deal with objects (i.e., constants), relations (i.e., predicates), and functions, the bootstrap generation is divided into three steps: generating the object hypothesis $H_O$, the relational hypothesis $H_R$, and the functional hypothesis $H_F$. Intuitively, the difficulties of generating these three hypotheses increase in order. While object hypothesis and relational hypothesis correspond more to the implicit and trivial statements in Figure 1, the functional hypotheses correspond more to the explicit and non-trivial statements.

### 4.1.1 Object hypothesis generation

In the first step, we generate hypotheses that describe the patterns of each object in the evidence (i.e., input or output). Such hypotheses may be relatively simple and specific to a particular object, but they are the starting points of bootstrap generation. Specifically, for each input $x^{(i)}$ (resp., output $y^{(i)}$), `ObjectHypothesisGenerator` identifies a list of object patterns $\widetilde{H}_x^{(i)}$ (resp., $\widetilde{H}_y^{(i)}$).

Since LLM may generate fallacious patterns, `ObjectHypothesisInconsistencyEliminator` is used to eliminate those patterns inconsistent with its relevant object from the pattern list. The filtered object hypothesis of $\widetilde{H}_x^{(i)}$ (resp., $\widetilde{H}_y^{(i)}$) is denoted by $H_x^{(i)}$ (resp., $H_y^{(i)}$). The object hypothesis $H_O$ contains all these filtered object patterns. Formally, $H_O \triangleq \bigwedge_{i=1}^{n} H_x^{(i)} \wedge H_y^{(i)}$. In addition to the LLM's hallucination, another benefit of eliminating inconsistency is ensuring $H_O \cup E$ is as consistent as possible. If each $H_x^{(i)}$ or $H_y^{(i)}$ is consistent (with its corresponding object), $H_O \cup E$ will be consistent, since $H_x^{(i)}$, $H_y^{(i)}$ are object-level (i.e., cannot contradict with $E$, which is function-level) and object-dependent (i.e., cannot contradict each other). The consistency of $H_O$ contributes to the consistency clause of Definition 3.1, detailed in Section 4.2.1.

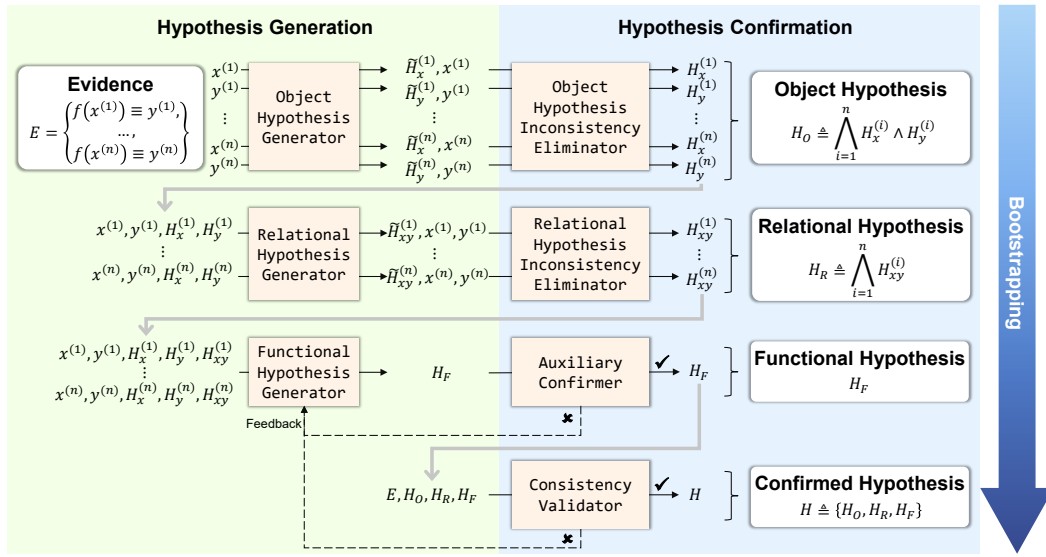

Figure 2: Overview of HypoBootstrap. The orange rectangles represent agents. See Figure 3 and Appendix C.2 for concrete examples.

### 4.1.2 Relational hypothesis generation

In the second step, we generate hypotheses for each input-output pair, which describe its relational patterns, by bootstrapping from the hypotheses for each object. Specifically, for each pair $(x^{(i)}, y^{(i)})$, RelationalHypothesisGenerator identifies a list of patterns $\widetilde{H}_{xy}^{(i)}$ between $x^{(i)}$ and $y^{(i)}$ based on $H_x^{(i)}, H_y^{(i)}, x^{(i)}, y^{(i)}$. For example, *the input and output are both number matrices of 5 columns and 5 rows*. At this step, the hypotheses are closer to functional hypotheses generated at the next step, since they already involve the relation between objects and are not merely object-specific. However, the relational hypotheses are still specific to a particular input-output pair and may be unable to form a function, i.e., it is probably impossible to determine the unique output for a certain input.

The relational hypothesis $H_R$ contains the relational patterns $H_{xy}^{(i)}$ filtered by RelationalHypothesisInconsistencyEliminator. Formally, $H_R \triangleq \bigwedge_{i=1}^{n} H_{xy}^{(i)}$. Similarly, if each $H_{xy}^{(i)}$ is consistent (with its corresponding input-output pair), $H_R \cup E$ will be consistent, since $H_{xy}^{(i)}$ are pair-dependent (i.e., cannot contradict each other). Besides, we assume $\{H_O, H_R\}$ is consistent enough, so is $\{H_O, H_R, E\}$ (justified in Appendix B.2).

### 4.1.3 Functional hypothesis generation

In the third step, bootstrapping from the object and relational hypotheses, we generate functional hypotheses for all input-output pairs, which describe the general transformation rule from input to output. Specifically, FunctionalHypothesisGenerator generates the functional hypothesis $H_F$ based on $\{H_O, H_R, E\}$. As illustrated in Figure 1, $H_F$ may be a natural-language hypothesis and imprecise. Combining with object and relational hypotheses, $H_O, H_R, H_F$ are expected to constitute a precise hypothesis and match the underlying formal hypothesis. We describe the confirmation of $H \triangleq \{H_O, H_R, H_F\}$ below.

### 4.2 Bootstrap confirmation

In this section, we bootstrap confirm $H = \{H_O, H_R, H_F\}$ using Definition 3.1. A crucial difference, in terms of hypothesis confirmation, against existing works [20, 25] is that, roughly speaking, existing works confirm $H_F$ which is typically imprecise as illustrated in Figure 1, but HypoBootstrap confirms

$H = \{H_O, H_R, H_F\}$ that is deliberately bootstrap-generated and is expected to be as precise as formal language.

Adapting Definition 3.1, defined in the first-order language, to LLMs, natural language-based models, is problematic. The pivotal issue is also LLM's hallucination, which makes strict confirmation rarely possible. Therefore, in the following subsections, we attempt to simplify Definition 3.1, with reasonable justification, to alleviate the impact of LLM's hallucination in our bootstrap confirmation. Note that it is our deliberately designed bootstrap generation strategy that permits most of the simplifications. Specifically, we validate the consistency clause, the confirmation clause, and the non-triviality clause of Definition 3.1 in Section 4.2.1, Section 4.2.2, and Section 4.2.3, respectively.

### 4.2.1 Consistency

The consistency of $H \cup E$, where $H = \{H_O, H_R, H_F\}$, is guaranteed in multiple parts. In Section 4.1.1 and Section 4.1.2, we have already ensured (as much as possible) the consistency of $\{H_O, H_R, E\}$. The consistency check between $H_F$ and $\{H_O, H_R, E\}$ is ensured by `ConsistencyValidator`. To reduce experimental cost, `ConsistencyValidator` is used after `AuxiliaryConfirmer` (see Section 4.2.2), since the latter costs less. If `AuxiliaryConfirmer` failed, `ConsistencyValidator` will not be used. [3]

The inconsistency elimination used in Section 4.1.1 and Section 4.1.2 vastly reduces the burden of `ConsistencyValidator` and the whole framework. If we did not eliminate inconsistency when generating $H_O$ and $H_R$, `ConsistencyValidator` would check the consistency among everything in $H \cup E$, i.e., $H_x^{(i)}, H_y^{(i)}, H_{xy}^{(i)}, H_F$, which would intuitively cause severe hallucination. Besides, also profiting from the inconsistency elimination, if `ConsistencyValidator` detect inconsistency in $H_F$, we can only regenerate $H_F$.

### 4.2.2 Confirmation

Recall that the confirmation clause of Definition 3.1 lacks an auxiliary confirmation method and HypoBootstrap uses an HD-like confirmation (i.e., similar to Definition 3.2), which we define here.

**Definition 4.1** (HypoBootstrap's Auxiliary Confirmation). Given a hypothesis $H$, the evidence $E$ confirms $H$ with respect to $H'$ if the program function, translated from $\{H, H'\}$ via LLMs, passes the unit test with $E$ as test cases.

This definition is a good approximation for the entailment clause of Definition 3.2 since it attempts to infer $E$ from $\{H, H'\}$ in a deductive way. The motivation of utilizing programs instead of LLMs is that LLMs are relatively not good at deduction with given rules, and can often produce correct programs if the rule is not too complex [20]. Definition 4.1 is in spirit similar to the hypothesis confirmation method used in [20, 25], in the case without $H'$ (i.e., $E$ confirms $H$, without respect to any auxiliary hypothesis). [4]

In Glymour's theory of bootstrap confirmation (Definition 3.1), checking the consistency clause of Definition 3.2 is superfluous since it is implied by the consistency clause of Definition 3.1. See Appendix A for detailed justification. What is lacking for Definition 3.2 is only the necessity clause.

Before discussing the necessity clause, we must first discuss the implementation details of the confirmation clause in Definition 3.1. According to the clause, we should find for each of $H_O, H_R, H_F$ a proper subset of $H = \{H_O, H_R, H_F\}$ as auxiliary hypotheses to help $E$ confirm it. In Hypo-Bootstrap, it is probably that verifying $H_O$ or $H_R$ according to the entailment clause of Definition 3.2, the core part of the confirmation clause, requires the help of $H_F$ since object hypothesis and relational hypothesis usually cannot imply function-generated evidence. Hence, the verification of the entailment clause in Definition 3.2 for $H_O$ or $H_R$ is implicitly accomplished if we verify $H_F$ for

---

[3]Appendix B.3 indicates that the prompt of `ConsistencyValidator` is much longer than `AuxiliaryConfirmer`. Our codebase provide real cases, showing that the prompt and response of `ConsistencyValidator` is much longer than `AuxiliaryConfirmer`. Hence, `ConsistencyValidator` is much more costly than `AuxiliaryConfirmer`.

[4]These works do not use Glymour's bootstrap confirmation. All they use is a method similar to Definition 4.1, which is only one part of our confirmation. See Appendix A.1 for more discussion.

the entailment clause with the help of $\{H_O, H_R\}$ (i.e., we have $\{H_O, H_R, H_F\} \vdash E$). Therefore, for simplicity, we only verify $H_F$ for the confirmation clause. [5]

Now we go back to the necessity clause in Definition 3.2. What we still need to check, according to the final method given above, is $\{H_O, H_R\} \nvdash E$. However, this check is unnecessary since $H_O$ and $H_R$ have little chance of completely describing the mapping $f$ that generates $E$.

It is worth recalling and emphasizing that although the final confirmation method, after quite a few simplifications, seems much simpler than the initial version, it is because of our elaborately designed bootstrap-generation method that we are reasonably permitted to do the simplification above.

### 4.2.3 Non-triviality

In the problem setting described in Section 3.1, checking the non-triviality clause of Definition 3.1 is not critical if the confirmation clause of Definition 3.1 is successfully verified via Definition 4.1. Specifically, successfully verified via Definition 4.1 indicates that for all $x \in \mathcal{X}$, there is a unique $y \in \mathcal{Y}$ such that $f(x) \equiv y$ is *true*, and that for any $y' \neq y$, $f(x) \equiv y'$ is *false*. [6] Analogous to Definition 4.1, we then have $H \vdash \neg f(x) \equiv y'$. By Definition 3.3, $f(x) \equiv y'$ is thus a possible, but not observed, evidence that *disconfirms* $H$.

### 4.3 Refinement

As depicted in Figure 2 and mentioned in Section 4.2.2, if $H_F$ or $H$ is not confirmed, we regenerate $H_F$ with error feedback. The feedback contains the wrong case from Definition 4.1 and its corresponding object hypotheses and relational hypotheses. We use $T$ to denote the number of refinement iterations ($T = 1$ indicates no refinement).

## 5 Experiments

We compare HypoBootstrap (HB) against Hypothesis Refinement (HR) [20] and MoC [13] to **demonstrate the effectiveness of both bootstrap generation and bootstrap confirmation**. Hypothesis Search (HS) [25] manually adds human priors into prompts; hence, it is not directly comparable to other methods, and cannot be easily extended to new tasks. Instead, we imitate HS to add human priors into agents to see whether the HypoBootstrap's performance can be further improved. We also conduct ablation studies and case studies to further demonstrate the effectiveness of HypoBootstrap's components.

### 5.1 Setup

Our experimental setup follows HR [20], detailed in this subsection. All experiments use GPT-4 [18]. We also include results on DeepSeek-V3 [3] in Appendix C.1. For a fair comparison, we ensure that the number of functional hypotheses generated across different methods remains the same, i.e., $N = 1$ in HR and $K = T$ in MoC. [7] This also ensures that the frequency of unit testing in training remains the same across all methods. [8] In addition, LLM's decoding temperature in HypoBootstrap is set to 0, i.e., greedy decoding.

### 5.1.1 Evaluation

Evaluation is conducted on a hold-out set of unobserved evidence, separated from the observed evidence used to infer rules. We use raw accuracy and task accuracy to compare inductive reasoning

---

[5]In practice, we found the LLM agent `AuxiliaryConfirmer` can generate a well-defined program solely based on $H_F$, and thus the helper hypothesis $\{H_O, H_R\}$ in Definition 4.1 (corresponding to the entailment clause) are unnecessary. Generating the program additionally based on $\{H_O, H_R\}$ may further increase the correctness, but may augment hallucination due to the long context. We choose to ignore them to reduce experimental cost.

[6]The only exception is that the program function involves randomness, preventing from producing outputs uniquely. We did not observe such case in experiments.

[7]MoC doesn't involve iterations. See their original paper for the definition of $N$ and $K$.

[8]More precisely, HR and HB may have one less unit test since not necessary in the final refinement iteration.

Table 1: Main results. $T$ refers to the number of refinement iterations. $K = T$ for MoC.

| T | Met. | ACRE | | List Fns | | MiniSCAN | | MiniARC | | Avg. | |
|---|---|---|---|---|---|---|---|---|---|---|---|
| | | Raw | Task | Raw | Task | Raw | Task | Raw | Task | Raw | Task |
| 1 | HR | 78.3 | 45.0 | 51.6 | 42.4 | 77.0 | 46.0 | 5.9 | 3.8 | 53.2 | 34.3 |
| | MoC | 74.0 | 34.0 | 53.0 | 43.2 | 60.4 | 21.0 | 6.7 | 4.6 | 48.5(-4.7) | 25.7(-8.6) |
| | $\widetilde{\text{HB}}$ | **84.8** | **57.0** | 53.5 | 42.0 | 92.9 | 75.0 | 9.5 | 4.6 | 60.2(+7.0) | 44.7(+10.4) |
| | HB | **84.8** | **57.0** | **57.9** | **47.6** | **94.9** | **82.0** | **11.0** | **6.9** | **62.2(+9.0)** | **48.4(+14.1)** |
| 3 | HR | 77.8 | 47.0 | 61.7 | 52.8 | **98.2** | **95.0** | 10.1 | 6.9 | 62.0 | 50.4 |
| | MoC | 73.0 | 39.0 | 61.9 | 50.8 | 87.3 | 64.0 | 10.8 | **8.5** | 58.3(-3.7) | 40.6(-9.8) |
| | $\widetilde{\text{HB}}$ | 79.0 | **56.0** | 59.9 | 50.4 | 93.2 | 78.0 | **12.6** | 7.7 | 61.2(-0.8) | 48.0(-2.4) |
| | $\widetilde{\text{HB}}$* | **79.3** | **56.0** | 61.8 | 50.4 | 96.2 | 79.0 | 12.3 | 7.7 | 62.4(+0.4) | 48.3(-2.1) |
| | HB | 78.5 | 55.0 | 62.4 | **54.0** | 96.6 | 89.0 | 11.0 | 7.7 | 62.1(+0.1) | 51.4(+1.0) |
| | HB* | 79.0 | 55.0 | **64.3** | 53.6 | **98.2** | 89.0 | 12.1 | **8.5** | **63.4(+1.4)** | **51.5(+1.1)** |

Table 2: Ablation results on List Functions.

| Method | T=1 | | T=3 | |
|---|---|---|---|---|
| | Raw | Task | Raw | Task |
| HB | **57.9** | **47.6** | **62.4** | **54.0** |
| $w/o\ H_O$ | 57.2 | 45.6 | **62.4** | 52.0 |
| $w/o\ H_R$ | 53.4 | 44.0 | 60.5 | 51.6 |
| $\widetilde{\text{HB}}$ | 53.5 | 42.0 | 59.9 | 50.4 |

Table 3: Results with human prior.

| Method | MiniARC | |
|---|---|---|
| | Raw | Task |
| HB (T=1) | **11.0** | 6.9 |
| $w$ human prior | **11.0** | 7.7 |
| HB (T=3) | 11.0 | 7.7 |
| $w$ human prior | **12.6** | **9.2** |

methods. An input-output pair is considered accurately predicted by a hypothesis if the program function, translated from the evaluating hypothesis via LLMs, passes the unit test with this pair as the test case (similar to Definition 4.1). The *raw accuracy* of a task is the average accuracy over all input-output pairs, and the raw accuracy of a dataset is averaged over all tasks involved. The *task accuracy* of a dataset is the percentage of tasks whose input-output pairs are all accurately predicted.

### 5.1.2 Datasets

We use four inductive reasoning datasets with varying natures: causal induction, concept learning, grammar learning, and abstract reasoning. See Qiu et al. [20] and our codebase for examples and more details.

The Abstract Causal REasoning dataset (**ACRE**) [27] is a diagnostic benchmark for causal induction. The presence of certain objects triggers a machine, and the goal is to find which objects can trigger it. Sampled 100 tasks from the original dataset are used. Following Gendron et al. [5], instead of images, evidence is represented in language. **List Functions** dataset [21] is initially designed for psychological investigation of concept learning, which requires establishing a mapping from input lists to output lists. We use the original 250 tasks. **MiniSCAN** [12] requires the ability of sequence-to-sequence learning. As in HR, a quasi-synchronous context-free grammar [23] is used to represent sequence-to-sequence rules and pseudowords to represent inputs. We use 100 generated tasks. The Abstract Reasoning Corpus (ARC) [1] is an advanced benchmark for measuring general fluid intelligence. **MiniARC** [10] is a small-scale version of ARC, where inputs and outputs are 5x5 visual grids. The remaining 130 tasks after heuristically filtering are used. The visual grids are transformed into text data by mapping the grid cells to corresponding integers.

### 5.2 Main results

Table 1 shows the main results. $\widetilde{\text{HB}}$ removes all the inconsistency elimination (Section 4.1.1 & 4.1.2) and consistency validation (Section 4.2.1) from HypoBootstrap. The only improvement of $\widetilde{\text{HB}}$ from HR is generating object hypothesis and relational hypothesis before generating functional hypothesis. Experimental results show that, when $T = 1$, the bootstrap generation gains a 7.0 percentage point improvement on raw accuracy, averaged over all four datasets, and a 10.4 percentage

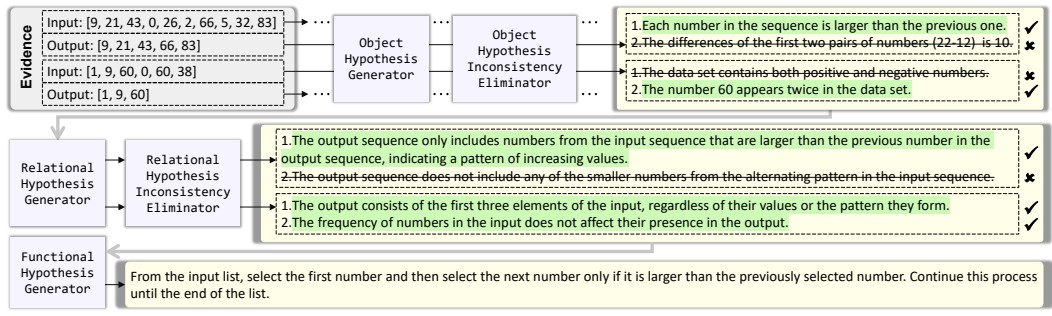

Figure 3: A real case from HypoBootstrap on List Functions.

point improvement on task accuracy, **significantly demonstrating the effectiveness of the bootstrap generation**. Besides, **the bootstrap confirmation can further improve performance**. When $T = 1$, HypoBootstrap gains an 9.0 & 14.1 percentage point improvement on raw accuracy & task accuracy, **significantly demonstrating the effectiveness of the whole framework**.

When using iterative refinement ($T = 3$), HypoBootstrap still outperforms HR on average. While HR, HB, $\widetilde{\text{HB}}$ stop refinement if the training task accuracy achieves one (without validating by `ConsistencyValidator`), and use results from the last iteration if no early stopping, HB* and $\widetilde{\text{HB}}$* use the results from the iteration with the best training raw accuracy. Results show that consistency-related agents are indispensable for iterative refinement. We also found that HypoBootstrap slightly underperforms HR on MiniSCAN when $T = 3$, caused by LLM's degraded long-context capability (Appendix C.3).

MoC is unstable across tasks. On ListFns and MiniARC, MoC outperforms HR or achieves comparable performance, which is in line with the results reported in MoC's paper. However, on ACRE and MiniSCAN, MoC shows poor performance. MoC is even significantly worse than HR in several metrics. The latter two tasks both require generating hypotheses under a specific format, and the correct hypotheses are usually highly composite, i.e., the output must be obtained by applying the rules in combination. This significantly differs from ListFns & MiniARC and makes ACRE & MiniSCAN a challenging benchmark. The comparison between the robust performance of HypoBootstrap and the unstable performance of MoC reveals the benefit of our framework.

### 5.3 Ablation study

The main results have already shown the individual effectiveness of bootstrap generation and bootstrap confirmation, respectively. In this section, we further analyze the effect of object hypothesis and relational hypothesis. Table 2 reveals the indispensability of both object hypothesis and relational hypothesis. However, the consistency-related agents in bootstrap confirmation is the most important according to Table 2.

### 5.4 Adding human prior

Similar to HS [25], we add human priors into agents' prompts, e.g., reasoning hints (Appendix B.4). Table 3 shows the improvement, and we believe more deliberately designed human priors (e.g., we don't use selected in-context examples as in HS) would further improve performance.

### 5.5 Case study

Figure 3 gives a real case of HypoBootstrap from our experiments. First, a list of object patterns is generated (in the light yellow box at the upper right) for each input or output. Inconsistent patterns are eliminated (struck through with a deletion line) while preserving the consistent ones (highlighted in green). The filtered object hypothesis is used to bootstrap-generate the relational hypotheses (within the light yellow box positioned centrally on the right side), a list of relational patterns of each input-output pair. As for object hypotheses, inconsistent relational patterns are filtered out.

Table 4: Token consumption for DeepSeek-V3. P for the number of prompt tokens and C for the number of completion tokens. $K = T$ for MoC. (Unit: thousand tokens)

| T | Met. | ACRE | | List Fns | | MiniSCAN | | MiniARC | | Avg. | |
|---|---|---|---|---|---|---|---|---|---|---|---|
| | | P | C | P | C | P | C | P | C | P | C |
| 1 | HR | **22** | **6** | **161** | **34** | **43** | **29** | 622 | 120 | 212 | **47** |
| | MoC | 46 | 24 | 209 | 67 | 306 | 417 | **152** | **50** | **178** | 140 |
| | HB | 401 | 69 | 2464 | 664 | 1201 | 245 | 991 | 283 | 1264 | 315 |
| 3 | HR | **56** | **10** | 539 | **76** | **331** | **77** | 1064 | 208 | 498 | **93** |
| | MoC | 98 | 23 | **439** | 82 | 454 | 464 | **317** | **90** | **327** | 165 |
| | HB* | 572 | 82 | 3242 | 840 | 1642 | 305 | 1616 | 407 | 1768 | 409 |

Finally, based on the object and relational hypotheses above, the functional hypothesis is bootstrap-generated for all input-output pairs. More importantly, compared with HR [20], whose generated functional hypothesis (i.e., rule) in this case is *removing all even numbers and zero from the list*, our method successfully generates the functional hypothesis since the pre-generated object and relational hypotheses automatically provide hints. See Appendix C.2 for more case study.

## 6 Limitations

Although HypoBootstrap achieves the best results on inductive reasoning, as shown in Table 1, it still cannot completely solve LLM's hallucination problem in inductive reasoning. For example, Appendix C.3 shows a wrongly generated rule after 3 refinement iterations. The main reason is that LLMs cannot strictly adhere to formal logic.

Another limitation is that HypoBootstrap has a higher inference cost compared to baselines. Table 4 shows the token consumption of each method. On average, compared with HR & MoC, HypoBootstrap consumes 6 & 7 times the prompt tokens and 7 & 2 times the completion tokens for 1 iteration, and 3 & 5 times the prompt tokens and 4 & 2 times the completion tokens for 3 iterations. We argue that the stable performance gain well justify the token consumption. In addition, HypoBoostrap has decreasing marginal cost in terms of refinement iterations, since only $H_F$ is regenerated in each refinement loop and the agents related to $H_O, H_R$ are only used once at the very beginning.

Besides, HypoBoostrap and all baselines require a way to verify generated hypotheses, e.g., unit test. Although this is available in research benchmarks, verification is not always available in real-world scenario, such as open-ended text-based induction. In scientific research, which is one of the largest applications of inductive reasoning, hypothesis verification may be costly due to expensive experimental instruments. Hence, reducing the need for verification is worth exploring.

## 7 Conclusion

LLM-based inductive reasoning allows direct reasoning in natural languages, without the need to convert into a specific formal language. To alleviate hallucinations and provide systematic guidance for LLM-based inductive reasoning, we propose HypoBootstrap, a framework integrating hypothesis bootstrap generation and hypothesis bootstrap confirmation. Experimental results on four different scenarios, including causal induction, concept learning, grammar learning, and abstract reasoning, demonstrate the effectiveness of both bootstrap generation and bootstrap confirmation. However, HypoBootstrap still cannot completely solve LLM's hallucination problem in inductive reasoning, though it achieves the best results. To this end, we emphasize that using HypoBootstrap in applications must consider the potential fallacious rules it produces.

## Acknowledgments and Disclosure of Funding

This work was supported by the National Natural Science Foundation of China (No. U2433212), in part by the Beijing Municipal Science and Technology Program (No. Z221100007122003), in

part by the Fundamental Research Funds for the Central Universities, and in part by the State Key Laboratory of Complex & Critical Software Environment.

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

## A  Glymour's theory of bootstrap confirmation

In this section, we introduce the first-order formalization of Glymour's theory of bootstrap confirmation introduced by Douven and Meijs [4] and justify how to transform it to Definition 3.1 in the case of using hypothetico-deductive confirmation (Definition 3.2) & disconfirmation (Definition 3.3) in the HypoBootstrap framework. Since HypoBootstrap uses a hypothetico-deductivism-like confirmation (Definition 4.1) & disconfirmation (Section 4.2.3), using Definition 3.1 is a rational approximation.

**Definition A.1** (Douven and Meijs [4]'s Formalization of Glymour's Theory of Bootstrap Confirmation). Given a hypothesis $H = \{H_1, H_2, \ldots, H_n\}$, evidence $E$ bootstrap-confirms $H$ if

1. $H \cup E \nvdash \bot$

2. for each $H_i$, there exists $H' \subsetneq H$ s.t. $H_i \notin H'$ and

   (a) $E$ *confirms* $H_i$ with respect to $H'$ ; and
   (b) there is possible evidence $E'$, but not actually observed, such that $E'$ *disconfirms* $H_i$ with respect to $H'$

3. for each $H_i$, there is no $H'' \subseteq H$ such that $E$ *disconfirms* $H_i$ with respect to $H''$.

The *confirm* & *disconfirm* refer to an auxiliary (non-bootstrap) confirmation & disconfirmation method. Glymour's original theory tends to be neutral across auxiliary (non-bootstrap) confirmation methods [6]. We select hypothetico-deductive confirmation & disconfirmation, where evidence confirms a hypothesis if the hypothesis can deductively entail the evidence, since it is one of the popular confirmation methods in scientific research. The formalization of hypothetico-deductive confirmation & disconfirmation has been given in Definition 3.2 & 3.3. In Definition 3.2 & 3.3, if $E$ serves as deduction consequence, it is understood as the conjunction of its elements. For example, $H \cup H' \vdash E$ is identical to $H \cup H' \vdash \bigwedge_{i=1}^{n} f(x^{(i)}) \equiv y^{(i)}$.

Since Glymour's bootstrap confirmation often seems circular at first sight, it is worth mentioning the significance of the clause 2.(b) of Definition A.1, which corresponds to the non-triviality clause of Definition 3.1. This clause makes the Definition A.1 not a circular confirmation. For a discussion, see Comment (6) for Definition 2.1 in Douven and Meijs [4]'s work.

Now we specify how to transform Definition A.1 to Definition 3.1. Douven and Meijs [4] indicates that using hypothetico-deductive confirmation can eliminate the third clause of Definition A.1 since it is entirely redundant. See Comment (3) for Definition 2.1 in Douven and Meijs [4]'s work for a proof. Hence, we only consider clauses 1, 2.(a), and 2.(b), which, respectively, correspond to the consistency clause, confirmation clause, and non-triviality clause of Definition 3.1.

**Simplification of non-triviality clause**   The non-triviality clause of Definition 3.1 is independent of $H'$ and different from the clause 2.(b). The non-triviality clause is a sufficient condition of the clause 2.(b) when the clause 2.(a) holds. We do such simplification because separately determining $E'$ for object hypothesis, relational hypothesis, and functional hypothesis are difficult and costly for LLMs. To see the sufficiency, recall we consider a HD disconfirmation (Definition 3.3), where checking $E'$ *disconfirms* $H$ is to check $H \vdash \neg E'$ (the consistency of $H$ is checked by the clause 1). Similarly, checking $E'$ *disconfirms* $H_i$ *with respect to* $H'$ is to check $H_i \cup H' \vdash \neg E'$. By these definitions, if $E'$ disconfirms $H$, then for each $H_i$, $E'$ disconfirms $H_i$ with respect to $H''' = H \setminus H_i$. Hence, $E'$ is a possible evidence that disconfirms any hypothesis. On the other hand, according to the clause 2.(a), we already have $H_i \cup H' \vdash E$ for some $H'$. By the monoticity of first-order logic and since $H' \subset H'''$, we have $H_i \cup H''' \vdash E$. If we set $H'$ to $H'''$, then the clause 2.(a) and 2.(b) both hold.

---

**Algorithm 1** HypoBootstrap

---

1: **input:** evidence $E = \{f(x^{(1)}) \equiv y^{(1)}, \ldots, f(x^{(n)}) \equiv y^{(n)}\}$, maximum iteration of refinement $T$
2: **output:** functional hypothesis $H_F$
3: /* Object Hypothesis */
4: **for** $i = 1$ **to** $n$ **do**
5:   $\widetilde{H}_x^{(i)} \leftarrow \texttt{ObjectHypothesisGenerator}(x^{(i)})$
6:   $\widetilde{H}_y^{(i)} \leftarrow \texttt{ObjectHypothesisGenerator}(y^{(i)})$
7:   $H_x^{(i)} \leftarrow \texttt{ObjectHypothesisInconsistencyEliminator}(\widetilde{H}_x^{(i)}, x^{(i)})$
8:   $H_y^{(i)} \leftarrow \texttt{ObjectHypothesisInconsistencyEliminator}(\widetilde{H}_y^{(i)}, y^{(i)})$
9: **end for**
10: $H_O \leftarrow \bigwedge_{i=1}^{n} H_x^{(i)} \wedge H_y^{(i)}$
11: /* Relational Hypothesis */
12: **for** $i = 1$ **to** $n$ **do**
13:   $\widetilde{H}_{xy}^{(i)} \leftarrow \texttt{RelationalHypothesisGenerator}(H_x^{(i)}, H_y^{(i)}, x^{(i)}, y^{(i)})$
14:   $H_{xy}^{(i)} \leftarrow \texttt{RelationalHypothesisInconsistencyEliminator}(\widetilde{H}_{xy}^{(i)}, x^{(i)}, y^{(i)})$
15: **end for**
16: $H_R \leftarrow \bigwedge_{i=1}^{n} H_{xy}^{(i)}$
17: /* Functional Hypothesis */
18: $E_{\text{wrong}} = \{\}$
19: **repeat**
20:   **if** it is the first iteration **then**
21:    $H_F \leftarrow \texttt{FunctionalHypothesisGenerator}(E, H_O, H_R)$
22:   **else**
23:    $H_F \leftarrow \texttt{FunctionalHypothesisGenerator}(E, H_O, H_R, H_F, E_{\text{wrong}})$
24:   **end if**
25:   $E_{\text{wrong}} \leftarrow \texttt{AuxiliaryConfirmer}(H_F, E)$
26:   **if** $E_{\text{wrong}}$ is not empty **then**
27:    **continue**
28:   **end if**
29:   **if** $\texttt{ConsistencyValidator}(E, H_O, H_R, H_F)$ **then**
30:    **break**
31:   **end if**
32: **until** reach the $T$-th iteration

---

**Consistency clause of Definition 3.2 is superfluous.** In the context of Definition 3.1, only the confirmation clause uses an auxiliary confirmation method, and the hypotheses in the confirmation clause (i.e., $H_i, H'$) are element or subset of the whole hypothesis to be bootstrap-confirmed (i.e., $H$). The consistency clause in Definition 3.1 already verifies $H \cup E \nvdash \bot$, which is accomplished in Section 4.2.1. By the monotonicity, if $\{H_i\} \cup H' \vdash \bot$, then $H \cup E \vdash \bot$. Hence, we don't need to check the consistency clause of Definition 3.2.

## A.1 Relation to existing works

The confirmation used by Hypothesis Refinement [20], Hypothesis Search [25], and MoC [13] is Definition 4.1. This is only one part of our confirmation since they only generate functional hypothesis and do not generate object hypothesis and relational hypothesis.

HypoBootstrap bootstrap-generates functional hypothesis based on object hypothesis and relational hypothesis, and all these hypotheses together form a complete hypothesis for the transformation rule. Consequently, in addition to using Definition 4.1, which is accomplished by `AuxiliaryConfirmer`, HypoBootstrap also uses `ObjectHypothesisInconsistencyEliminator`, `RelationalHypothesisInconsistencyEliminator`, and `ConsistencyValidator` to confirm the hypothesis.

# B HypoBootstrap details

This section shows more details of HypoBootstrap, including the pseudo-code (B.1), justification for relational hypothesis generation (B.2), agents' details (B.3), and prompt with human prior (B.4).

## B.1 Pseudo-code

The pseudo-code of HypoBootstrap is given in Algorithm 1. Algorithm 1 stops if all training evidence can pass `AuxiliaryConfirmer` (i.e., $E_{\text{wrong}}$ is empty) and `ConsistencyValidator`. Slightly different from Algorithm 1, the variant HB* selects the functional hypothesis from the iteration with the best raw accuracy on observed evidence.

## B.2 Justification for relational hypothesis generation

One may question the assumption that $\{H_O, H_R\}$ is consistent enough since we do not eliminate the inconsistency between $H_x^{(i)}, H_y^{(i)}$ and $H_{xy}^{(i)}$. We argue that this inconsistency is less severe since $H_{xy}^{(i)}$ is generated based on $H_x^{(i)}, H_y^{(i)}$, and $H_x^{(i)}, H_y^{(i)}, H_{xy}^{(i)}$ are, respectively, already consistent with $E$. To reduce the cost, we do not explicitly eliminate the inconsistency between $H_x^{(i)}, H_y^{(i)}$ and $H_{xy}^{(i)}$.

We also have two extra comments on inconsistency elimination, which is accomplished by `ObjectHypothesisInconsistencyEliminator` and `RelationalHypothesisInconsistencyEliminator`. First, these two agents remove inconsistent object patterns from the pattern list or remove inconsistent relational patterns from the relational pattern list. Filtering in this way does not increase inconsistency, even if they may accidentally remove consistent patterns. Second, eliminating patterns in $H_O, H_R$ may reduce the hallucination of `FunctionalHypothesisGenerator` by decreasing the length of the input context, since LLMs are usually error-prone to long context.

## B.3 Agents

The agents' prompts used for GPT-4 and List Functions dataset are shown in Table 5, where the auxiliary prompt templates are shown in Table 6. The agents for other datasets are similar, and the differences arise from different task characteristics. For example, the `AuxiliaryConfirmer` for MiniSCAN and ACRE is accomplished via symbolic validators since their functional hypotheses are organized in formal structures. Refer to our codebase for more details on all datasets and all LLMs. Below, we describe the entire procedure with List Functions task.

`ObjectHypothesisGenerator` generates a list of object hypotheses of given data, where `{object}` refers to the input or output of each task. The object hypotheses are then fed into `ObjectHypothesisInconsistencyEliminator`, which identifies inconsistent hypotheses, generates their numbers, and filters them out based on the generated numbers.

Similarly, `RelationalHypothesisGenerator` generates a list of relational hypotheses between a given input and output according to their object hypothesis. `RelationalHypothesisInconsistencyEliminator` identifies inconsistent hypotheses, generates their numbers, and filters them out based on the generated numbers.

The `FunctionalHypothesisGenerator` generates the functional hypothesis in two modes: without feedback and with feedback. The generation without feedback takes all the input-output pairs as examples, along with all the consistent object hypotheses and relational hypotheses generated in previous steps. The generation with feedback leverages the previous functional hypothesis, unit test feedback, and the corresponding object hypotheses and relational hypotheses of the wrong examples.

The `AuxiliaryConfirmer` evaluates the functional hypothesis by unit tests. After passing the unit test on observed evidence, the `ConsistencyValidator` checks whether the rule is logically consistent with the input-output pairs and their object hypothesis and relational hypothesis.

Table 5: Prompts used for List Functions.

| Agent | Prompt |
|---|---|
| Object Hypothesis Generator | Observe the following data, which consists of integers.
Systematically analyze its patterns.
Provide up to three most important patterns in your analysis.
Data: {object}

Please format your analysis as follows:

1.
2.
3. |
| Object Hypothesis Inconsistency Eliminator | Observe the data and its patterns below.

Data: {object}

Patterns:
{object_hypothesis}

Which patterns are logically inconsistent with the data and other patterns?

Please respond with pattern numbers as follows:
Inconsistent patterns: <number>, <number>, <number>, ... |
| Relational Hypothesis Generator | Observe the input-output pair and its patterns below.
Systematically analyze the correlations between the input and the output.
Provide up to three most important correlations in your analysis.
Input: {input}
Output: {output}

Patterns of the input:
{input_object_hypothesis}

Patterns of the output:
{output_object_hypothesis}

Please format your analysis as follows:

1.
2.
3. |
| Relational Hypothesis Inconsistency Eliminator | Observe the input-output pair and the correlations between the input and the output given below.

Input: {input}
Output: {output}

Correlations:
{relational_hypothesis}

Which correlations are logically inconsistent with the input-output pair and other correlations?

Please respond with correlation numbers as follows:
Inconsistent correlations: <number>, <number>, <number>, ... |

| Agent | Prompt |
|---|---|
| Functional Hypothesis Generator (*w/o* feedback) | Generate a transformation rule that converts each of the given input lists into their corresponding output lists.

{examples}

Please format your rule as follows:
Rule: <Your rule>

Below, we collect the patterns and correlations of several input-output pairs mentioned above.

---

{examples_with_object_and_relational_hypothesis}

--- |
| Functional Hypothesis Generator (*w* feedback) | Based on your previous rule and the feedback, generate a revised transformation rule that converts each of the given input lists into their corresponding output lists.

Your previous rule: {functional_hypothesis}

Feedback:
{feedback}

Please format your revised rule as follows:
Rule: <Your rule>

Below, we collect the patterns and correlations of several input-output pairs mentioned above.

---

{feedback_with_object_and_relational_hypothesis}

--- |
| Auxiliary Confirmer | Write a Python function `fn` for the following rule, where the input must be a list of integers and the output must also be a list of integers.

Rule: {functional_hypothesis} |
| Consistency Validator | Is the target statement logically consistent with the input-output pairs and their patterns and correlations?

Target statement: {functional_hypothesis}

---

{examples_with_object_and_relational_hypothesis}

---

Please respond with Yes or No. |

Table 6: Auxiliary prompt templates for List Functions.

| Template | Prompt |
|---|---|
| example_template | Input: {input}
Output: {output} |
| example_with_object_and
_relational_hypothesis_template | Pair {number}
Input: {input}
Output: {output}

Patterns of the input:
{input_object_hypothesis}

Patterns of the output:
{output_object_hypothesis}

Correlations between input and output:
{relational_hypothesis} |
| feedback_template | Input: {input}
Expected output: {output}
Predicted output: {prediction} |
| feedback_with_object_and
_relational_hypothesis_template | Input: {input}
Expected output: {output}
Predicted output: {prediction}

Patterns of the input:
{input_object_hypothesis}

Patterns of the output:
{output_object_hypothesis}

Correlations between input and expected output:
{relational_hypothesis} |

## B.4 Prompt with human prior

The prompt with human prior for MiniARC (used in Section 5.4) is shown in Table 7.

Table 8: Results on DeepSeek-V3.

| T | Met. | ACRE Raw | ACRE Task | List Fns Raw | List Fns Task | MiniSCAN Raw | MiniSCAN Task | MiniARC Raw | MiniARC Task | Avg. Raw | Avg. Task |
|---|------|------|------|------|------|------|------|------|------|------|------|
|   | HR   | 72.0 | 37.0 | 44.4 | 32.8 | 25.8 | 21.0 | 9.7 | 6.9 | 38.0 | 24.4 |
| 1 | MoC  | 70.5 | 34.0 | 52.5 | 40.8 | 53.8 | 44.0 | 9.2 | 6.2 | 46.5(+8.5) | 31.3(+6.9) |
|   | HB   | **75.8** | **48.0** | **55.6** | **42.0** | **69.8** | **47.0** | **16.7** | **10.0** | **54.5(+16.5)** | **36.8(+12.4)** |
|   | HR   | 77.3 | **53.0** | 52.2 | 41.6 | 34.3 | 24.0 | 16.2 | 11.5 | 45.0 | 32.5 |
| 3 | MoC  | 73.5 | 43.0 | 59.2 | 48.4 | **80.0** | **61.0** | 14.9 | 11.5 | 56.9(+11.9) | 41(+8.5) |
|   | HB*  | **77.5** | **53.0** | **62.8** | **48.8** | 74.0 | 53.0 | **21.8** | **14.6** | **59.0(+14.0)** | **42.4(+9.9)** |

Table 7: Prompt with human prior for MiniARC.

| Agent | Prompt |
|-------|--------|
| Functional Hypothesis Generator (*w/o* feedback) | Generate a transformation rule that converts each of the given input matrices into their corresponding output matrices. Each input and output is a grid of numbers representing a 5x5 visual grid. The transformation rule may involve counting or sorting objects (e.g. sorting by size), comparing numbers (e.g. which shape or symbol appears the most? Which is the largest object? Which objects are the same size?), or repeating a pattern for a fixed number of time. There are other concepts that may be relevant. - Lines, rectangular shapes - Flipping objects - Rotating objects - Translating objects - Shape upscaling or downscaling, elastic distortions. - Containing / being contained / being inside or outside of a perimeter. - Drawing lines, connecting points, orthogonal projections. - Copying, repeating objects. - Pushing objects to a side  {examples}  Please format your rule as follows: Rule: <Your rule>  Below, we collect the patterns and correlations of several input-output pairs mentioned above.  ---  {examples_with_object_and_relational_hypothesis}  --- |

# C   More experimental results

This section includes DeepSeek-V3 results (C.1), more case study (C.2), and failure analysis (C.3).

## C.1   Results on DeepSeek-V3

In addition to GPT-4, we include experimental results on DeepSeek-V3 in Table 8. We can draw similar conclusions as from GPT-4 results.

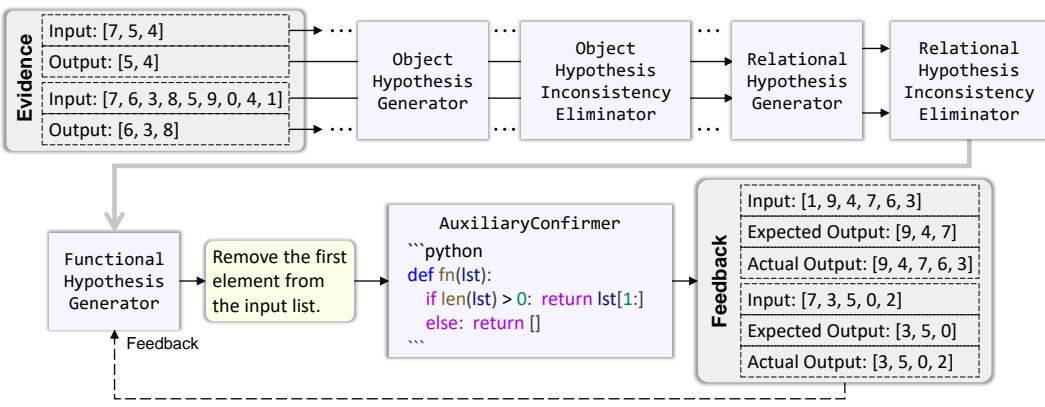

Figure 4: A case for HypoBootstrap's `AuxiliaryConfirmer` from List Functions.

## C.2 More case study

Figure 4 shows a case of HypoBootstrap's `AuxiliaryConfirmer` from List Functions. The functional hypotheses overfit to certain observed evidence and are inconsistent with other observed evidence. The partially correct generated functional hypotheses were evaluated by `AuxiliaryConfirmer` but did not pass the unit test with observed evidence. The feedback containing the failure cases of observed evidence is collected for the `FunctionalHypothesisGenerator` to generate revised functional hypotheses. Without `AuxiliaryConfirmer`, HypoBootstrap is incapable of rectifying incorrect functional hypotheses since it cannot identify the failure cases from observed evidence.

More real cases can be found in our codebase.

## C.3 Failure analysis

An example of failure case when $T = 3$ from MiniSCAN shows that our method requires robust long-context comprehension capabilities, which current LLMs still struggle to achieve. This results in slightly inferior performance compared to HR [20] on MiniSCAN when $T = 3$. As shown in this example, the generated rules did not follow the prompt's format due to the long context.

---

**User (Prompt for Functional Hypothesis Generation)**

```
Generate grammar rules that map the following inputs to their corresponding
outputs. Your grammar rules should follow the format "<input> -> <output>". Use
the prefix "##" to denote a nonterminal symbol. For instance, "##A twice -> ##A ##
A", "##A swap ##B -> ##B ##A". The left-hand side cannot contain repetitive or
adjacent nonterminal symbols; i.e., rules like "##A ##A -> ##A twice" or "##A ##B
-> ##B ##A" are not allowed. Ensure that the number of unique nonterminal symbols
on the left-hand side matches that on the right-hand side in your rules.
For each rule, assign an integer as its priority. A higher priority indicates that
 the rule should be considered first when generating parses. Ensure that
unnecessary colors are replaced by nonterminal symbols and that each rule has
pseudoword(s) in the left-hand side.
Try to make your rules as minimal as possible.

Input: "mccluiss"
Output: "RED"
Input: "szeaurt"
Output: "BLUE"
Input: "fioct"
Output: "GREEN"
Input: "cyclaiops"
Output: "YELLOW"
Input: "szeaurt bingly"
Output: "BLUE BLUE BLUE"
Input: "mccluiss bingly"
Output: "RED RED RED"
Input: "szeaurt staioff fioct"
```

```
Output: "BLUE GREEN BLUE"
Input: "fioct staioff mccluiss"
Output: "GREEN RED GREEN"
Input: "szeaurt biouck fioct"
Output: "GREEN BLUE"
Input: "mccluiss biouck szeaurt"
Output: "BLUE RED"
Input: "szeaurt bingly biouck fioct"
Output: "GREEN BLUE BLUE BLUE"
Input: "fioct biouck mccluiss staioff szeaurt"
Output: "RED BLUE RED GREEN"
Input: "szeaurt biouck fioct bingly"
Output: "GREEN GREEN GREEN BLUE"
Input: "fioct staioff mccluiss biouck szeaurt"
Output: "BLUE GREEN RED GREEN"
```

Please format your rules as follows:

```
Rule 1: <input> -> <output>
Priority 1: <Your priority>
...
```

(Below, we collect the patterns and correlations of several input-output pairs mentioned above.)

---

Pair 5
Input: "szeaurt bingly"
Output: "BLUE BLUE BLUE"

Patterns of the input:
1. The data consists of two separate pseudowords: "szeaurt" and "bingly".
2. Both pseudowords are single words, not sentences or phrases.
3. There is a space separating the two pseudowords.

Patterns of the output:
1. The data consists of a single repeated element, which is a color.
2. The color repeated in the data is "BLUE".
3. The repetition occurs three times.

Correlations between input and output:
1. The output does not directly correspond to the individual pseudowords in the input, suggesting that the transformation is not a simple one-to-one mapping.
2. The output is a repetition of a single color word, "BLUE", three times, regardless of the number of words in the input.
3. The specific color in the output does not seem to be influenced by the specific pseudowords in the input, suggesting that the transformation might be independent of the specific input content.

---

Pair 6
Input: "mccluiss bingly"
Output: "RED RED RED"

Patterns of the input:
1. The data consists of pseudowords, not colors.
2. There are two distinct pseudowords separated by a space.
3. Each pseudoword is a single, unbroken string of letters, suggesting they may represent individual units or entities.

Patterns of the output:
1. The data consists of a single repeated element, which is a color.
2. The color repeated in the data is "RED".
3. The repetition occurs three times.

Correlations between input and output:
1. The input data, regardless of its content or structure, consistently results in the output of the color "RED".
2. The number of words in the input does not affect the number of times "RED" is repeated in the output, which is always three.
3. The specific pseudowords used in the input do not seem to influence the color output, suggesting that the system does not differentiate between different inputs.

---

```
Pair 7
Input: "szeaurt staioff fioct"
Output: "BLUE GREEN BLUE"

Patterns of the input:
1. The data consists of pseudowords, not colors.
2. Each pseudoword is seven letters long.
3. All pseudowords are separated by a space.

Patterns of the output:
1. The data consists of colors, not pseudowords.
2. The sequence of colors is repetitive, with "BLUE" appearing twice and "GREEN"
once.
3. The pattern of the data is "BLUE", "GREEN", "BLUE".

Correlations between input and output:
1. Each pseudoword in the input corresponds to a color in the output. The first
and third pseudowords correspond to "BLUE", while the second pseudoword
corresponds to "GREEN".
2. The pseudowords do not seem to have a direct phonetic or alphabetic correlation
 to the colors they represent. The correlation might be based on a predefined or
coded system.
3. The repetition of the color "BLUE" in the output corresponds to the repetition
of the first pseudoword in the input, suggesting a consistent one-to-one mapping
between specific pseudowords and colors.

---

Pair 8
Input: "fioct staioff mccluiss"
Output: "GREEN RED GREEN"

Patterns of the input:
1. The data consists of pseudowords, not colors.
2. Each pseudoword is separated by a space.
3. The pseudowords vary in length, ranging from 5 to 8 characters.

Patterns of the output:
1. The data consists of colors, not pseudowords.
2. The sequence of colors is repetitive, with "GREEN" appearing twice and "RED"
appearing once in between.
3. The data does not show any progression or gradient in terms of color shades or
hues.

Correlations between input and output:
1. Each pseudoword in the input corresponds to a color in the output, suggesting a
 mapping between pseudowords and colors.
2. The pseudoword "fioct" and "mccluiss" both map to the color "GREEN", indicating
 that different pseudowords can correspond to the same color.
3. The pseudoword "staioff" maps to the color "RED", showing that each pseudoword
is associated with a specific color.

---

Pair 9
Input: "szeaurt biouck fioct"
Output: "GREEN BLUE"

Patterns of the input:
1. The data consists of pseudowords, not colors.
2. Each pseudoword is separated by a space.
3. There are three pseudowords in the data.

Patterns of the output:
1. The data consists of two distinct elements separated by a space.
2. Both elements in the data are colors.
3. The colors represented are primary (blue) and secondary (green) colors.

Correlations between input and output:
1. Each pseudoword in the input corresponds to a color in the output. The first
and second pseudowords correspond to "GREEN" and "BLUE" respectively. The third
pseudoword does not have a corresponding color in the output.
2. The pseudowords in the input do not seem to have a direct phonetic or
alphabetic correlation to the colors in the output. The correlation might be based
 on a hidden or coded system.

---
```

```
Pair 10
Input: "mccluiss biouck szeaurt"
Output: "BLUE RED"

Patterns of the input:
1. The data consists of pseudowords, not colors.
2. All pseudowords are single words, not separated by hyphens or other punctuation
.
3. Each pseudoword is unique, with no repetition of the same pseudoword.

Patterns of the output:
1. The data consists of two distinct elements separated by a space.
2. Both elements in the data are colors, specifically "BLUE" and "RED".
3. The data does not contain any pseudowords.

Correlations between input and output:
1. The output colors "BLUE" and "RED" are derived from the letters present in the
pseudowords of the input. The pseudoword "biouck" contains all the letters for "
BLUE" and "mccluiss" contains all the letters for "RED".
2. The order of the colors in the output does not correspond to the order of the
pseudowords in the input. The first pseudoword "mccluiss" corresponds to the
second color "RED", and the second pseudoword "biouck" corresponds to the first
color "BLUE".
3. The pseudoword "szeaurt" does not contribute to the output, suggesting that not
 all input pseudowords are used to generate the output.

---

Pair 11
Input: "szeaurt bingly biouck fioct"
Output: "GREEN BLUE BLUE BLUE"

Patterns of the input:
1. The data consists entirely of pseudowords, not colors.
2. Each pseudoword is separated by a space, indicating distinct units.
3. All pseudowords are of varying lengths, ranging from 5 to 7 letters.

Patterns of the output:
1. The data consists of colors.
2. The color "BLUE" is repeated three times.
3. The sequence starts with the color "GREEN" followed by three "BLUE".

Correlations between input and output:
1. Each pseudoword in the input corresponds to a color in the output, indicating a
 one-to-one mapping between pseudowords and colors.
2. The pseudoword "szeaurt" corresponds to the color "GREEN", while "bingly", "
biouck", and "fioct" all correspond to the color "BLUE".
3. The repetition of the color "BLUE" in the output suggests that different
pseudowords can correspond to the same color in the output.

---

Pair 12
Input: "fioct biouck mccluiss staioff szeaurt"
Output: "RED BLUE RED GREEN"

Patterns of the input:
1. All entries in the data set are pseudowords, not colors.
2. Each pseudoword in the data set consists of seven letters.
3. The pseudowords do not follow a consistent pattern in terms of vowel and
consonant placement.

Patterns of the output:
1. The color "RED" is repeated twice in the data set.
2. The sequence of colors does not follow a standard color spectrum or any
discernible pattern based on color properties.

Correlations between input and output:
1. The output colors seem to be determined by the third and fourth letters in each
 pseudoword of the input. If the third and fourth letters are 'oc', the output is
'RED'. If they are 'ou', the output is 'BLUE'. If they are 'ai', the output is '
GREEN'.
2. The number of pseudowords in the input matches the number of colors in the
output, suggesting a one-to-one correspondence between pseudowords and colors.
3. The order of the colors in the output matches the order of the pseudowords in
the input, suggesting that the sequence of pseudowords determines the sequence of
colors.
```

```
---

Pair 13
Input: "szeaurt biouck fioct bingly"
Output: "GREEN GREEN GREEN BLUE"

Patterns of the input:
1. The data consists of pseudowords, not colors.
2. Each pseudoword is separated by a space.
3. The pseudowords vary in length, ranging from 5 to 7 letters.

Patterns of the output:
1. The data primarily consists of the repetition of the word "GREEN" three times
consecutively.
2. The word "BLUE" appears only once and it is at the end of the sequence.
3. All data points are colors.

Correlations between input and output:
1. The first three pseudowords in the input correspond to the color "GREEN" in the
 output, while the last pseudoword corresponds to "BLUE". This suggests a one-to-
one mapping between pseudowords and colors.

---

Pair 14
Input: "fioct staioff mccluiss biouck szeaurt"
Output: "BLUE GREEN RED GREEN"

Patterns of the input:
1. The data consists entirely of pseudowords, not colors.
2. All pseudowords in the data set are composed of a combination of consonants and
 vowels.
3. The pseudowords vary in length, ranging from 5 to 8 characters.

Patterns of the output:
1. The data consists of colors, not pseudowords.
2. The colors are repeated, specifically "GREEN" appears twice.
3. The sequence of colors does not follow a standard color spectrum or rainbow
order.

Correlations between input and output:
1. Each pseudoword in the input corresponds to a color in the output. The
pseudoword "fioct" corresponds to "BLUE", "staioff" and "szeaurt" correspond to "
GREEN", and "mccluiss" corresponds to "RED".
2. The pseudowords do not seem to have a direct phonetic or spelling correlation
to the colors they represent. Instead, the correlation seems to be based on a
predefined mapping.
3. The order of the pseudowords in the input determines the order of the colors in
 the output. The first pseudoword corresponds to the first color, the second
pseudoword to the second color, and so on.

---
```

Assistant (Response from Functional Hypothesis Generation)

```
Based on the patterns and correlations identified, the following grammar rules can
 be proposed:

Rule 1: "mccluiss" -> "RED"
Priority: 1

Rule 2: "szeaurt" -> "BLUE"
Priority: 1

Rule 3: "fioct" -> "GREEN"
Priority: 1

Rule 4: "cyclaiops" -> "YELLOW"
Priority: 1

Rule 5: "##A bingly" -> "##A ##A ##A"
Priority: 2

Rule 6: "##A staioff ##B" -> "##A ##B ##A"
Priority: 3
```

```
Rule 7: "##A biouck ##B" -> "##B ##A"
Priority: 4

Rule 8: "##A bingly biouck ##B" -> "##B ##A ##A ##A"
Priority: 5

Rule 9: "##A biouck ##B staioff ##C" -> "##B ##A ##B ##C"
Priority: 6

Rule 10: "##A biouck ##B bingly" -> "##B ##B ##B ##A"
Priority: 7

Rule 11: "##A staioff ##B biouck ##C" -> "##C ##A ##B ##C"
Priority: 8

Note: The nonterminal symbol "##A", "##B", "##C" represent any pseudoword that can
 be replaced by a color according to the rules 1-4. The priority of the rules is
set based on the complexity of the rule, with more complex rules having higher
priority.
```

## D   Converting natural-language rules into first-order languages

This section gives examples to convert natural-language statements into first-order languages as in Figure 1. The natural-language statements are related to our used datasets.

**MiniARC**   *Fill the output grid with the number of non-zero elements in the input* can be converted into the following first-order statement

$$\bigwedge_{i=1,j=1}^{5,5} \texttt{elem}(y,i,j) \equiv g(x)$$

where $\texttt{elem}(y,i,j)$ is the element of the output $y$ in the $i$-th row and $j$-th column, $g(x)$ is a function that counts the number of non-zero elements in the input $x$.

**List Functions**   See Figure 1.

**ACRE**   *The red cylinder can trigger the machine, but the black cube cannot trigger the machine* can be converted into the following first-order statement

$$\texttt{trigger}(a) \wedge \neg\texttt{trigger}(b)$$

where $a$ refers to the object *red cylinder*, $b$ refers to the object *back cube*, and $\texttt{trigger}(c)$ asserts whether the object $c$ can trigger the machine.

**MiniSCAN**   *The symbol gsecua swaps its adjacent two words* can be converted into the following first-order statement

$$\texttt{elem}(x,i) \equiv \texttt{gsecua} \rightarrow \texttt{elem}(y,i-1) \equiv \texttt{elem}(x,i+1) \wedge \texttt{elem}(y,i+1) \equiv \texttt{elem}(x,i-1)$$

where $\texttt{elem}(x,i)$ is the $i$-th element of the input sequence $x$ and $y$ is the output sequence. This example is simplified since we don't consider the influence of other words in the sequence and retain the symbol gsecua in the output.

## E   Broader impacts

In real-world applications, our method can contribute to scientific hypothesis proposing, daily-life inductive reasoning, etc. If properly used, HypoBootstrap can enhance the inductive reasoning performance. However, we emphasize that using HypoBootstrap must consider the potential fallacious rules it produces and that one shouldn't completely believe in the results produced by HypoBootstrap.

