# OpenReview forum: "HypoBootstrap: A Bootstrapping Framework for Inductive Reasoning"
_NeurIPS.cc/2025/Conference — NeurIPS 2025 poster_

### Official Review · Reviewer_6r5u · 2025-07-01

**Clarity:** 3
**Significance:** 3
**Originality:** 3
**Rating:** 4
**Confidence:** 3

**Summary:**

This paper addresses the challenge of enhancing the inductive reasoning capabilities of Large Language Models (LLMs) while mitigating their propensity for hallucination. To this end, the authors propose HypoBootstrap, a novel framework that employs a structured, multi-stage process to generate and validate hypotheses. The generation process is progressive, beginning with the identification of simple Object Hypotheses concerning individual data points, followed by the establishment of Relational Hypotheses (HR​) between input-output pairs, and culminating in the general Functional Hypothesis (HF​). This pipeline is tightly integrated with a bootstrap confirmation mechanism, an adaptation of Glymour's confirmation theory from the philosophy of science. The role of this mechanism is to validate the entire set of hypotheses for logical consistency and correctness, for instance, by eliminating inconsistent statements and programmatically verifying the final hypothesis against the evidence. The authors provide a comprehensive empirical evaluation on four diverse inductive reasoning benchmarks, where HypoBootstrap is shown to significantly outperform existing methods.

**Questions:**

1. The framework seems to have a high computational overhead. Can you provide a more direct quantitative comparison of this cost against the baseline and discuss how the approach is expected to scale?

2. How robust is the framework to flawed or uninformative intermediate hypotheses (HO,HR)? Since the refinement loop only targets HF​, are there cases where feedback should instead revise the more foundational HO or HR ?

**Ethical Concerns:**

["NO or VERY MINOR ethics concerns only"]

**Final Justification:**

We thank the authors for their response, which we have carefully reviewed.  With my main concerns addressed, I’ll maintain my evaluation.

**Limitations:**

1. Quantify computational and environmental costs.
While Appendix D.1 reports per-module resource usage, the paper lacks a holistic comparison of total LLM calls, wall-clock time, API cost against baseline methods—and no analysis of how this overhead grows with dataset size or input length.

2. Clarify the framework’s scope and generality.
HypoBootstrap rests on converting hypotheses into executable unit tests. The authors should explicitly state this dependency and discuss its applicability (or breakdown) when such tests cannot be constructed—for example, in open-ended text-only inductive tasks—so readers understand where the method may fail or need adaptation.

**Paper Formatting Concerns:**

No major formatting issues were found.

**Quality:**

3

**Strengths And Weaknesses:**

Positive Aspects:
1. The work's primary strength is its originality in adapting Glymour's bootstrap confirmation theory from the philosophy of science to guide inductive reasoning in LLMs. This provides a novel, theoretically-grounded alternative to purely heuristic methods.
2. The proposed framework is technically robust. The multi-stage generation process is a logical and effective way to decompose the reasoning task. Furthermore, the experimental validation is rigorous, demonstrating significant performance gains on four diverse benchmarks and using insightful ablation studies to confirm the contribution of each component.
3. The paper is clearly written and well-presented. The authors effectively use figures to illustrate the framework's complex architecture and provide a compelling case study of its practical advantages.
Weaknesses：
1. The empirical results are reported without error bars or statistical significance tests. The authors note this is due to the high cost of replicating experiments on GPT-4.
2. The framework's multi-step process, involving numerous LLM agent calls, likely incurs a significant computational overhead compared to baseline methods. This trade-off is not explicitly analyzed in the main paper.
3. There is a minor inaccuracy in the submitted checklist regarding the declaration of LLM usage , which should be corrected to reflect that LLMs are a core component of the methodology.

---

> ### Author Rebuttal · Authors · 2025-07-31
>
> # Reviewer 6r5u
>
> We sincerely thank you for the insightful comments and constructive feedback, which are highly valuable in improving our manuscript.
> We also appreciate your recognition of the novelty, principled foundation, technically robustness, and high effectiveness of our method.
>
>
> In the following, we address each of your questions in detail, point by point.
>
>
> > **Question 1**: The framework seems to have a high computational overhead. Can you provide a more direct quantitative comparison of this cost against the baseline and discuss how the approach is expected to scale?
> >
> > **Weakness 5**: The framework's multi-step process, involving numerous LLM agent calls, likely incurs a significant computational overhead compared to baseline methods. This trade-off is not explicitly analyzed in the main paper.
> >
> > **Limitation 1**: Quantify computational and environmental costs. While Appendix D.1 reports per-module resource usage, the paper lacks a holistic comparison of total LLM calls, wall-clock time, API cost against baseline methods—and no analysis of how this overhead grows with dataset size or input length.
>
> A fair way to compare the computational overhead for LLMs is the token consumption.
> Here is the number of tokens consumed by all methods.
> (It's slightly simpler to count tokens on DeepSeek-V3, considering the limited rebuttal period. And other reviewers requeust us to conduct experiments on open-source models)
>
>
>
> [DeepSeek-V3; $T=1$] (P for prompt tokens; C for completion tokens)
> | Method | ACRE (P\C) | List Fns (P\C) | MiniSCAN (P\C) | MiniARC (P\C) |
> | - | - | - | - | - |
> | HR | 22\6 | 103\120 | 43\29 | 99\215 |
> | MoC | 46\24 | 209\67 | 80\72 | 152\50 |
> | HB (ours) | 400\69 | 2409\633 | 1201\245 | 991\283 |
>
>
>
> [DeepSeek-V3; $T=3$]
> | Method | ACRE (P\C) | List Fns (P\C) | MiniSCAN (P\C) | MiniARC (P\C) |
> | - | - | - | - | - |
> | HR | 56\10 | 477\352 | 331\77 | 565\532 |
> | MoC | 98\23 | 439\81 | 177\142 | 317\90 |
> | HB* (ours) | 572\82 | 3121\745 | 1642\305 | 1616\407 |
>
>
>
> Our framework does consume about 1~10 times tokens than baseline methods.
> However, this is not surprising since
> - **Our framework is much more robust across models and datasets than baselines**. See our response to Reviewer tLyE for results on DeepSeek-V3 and MoC (if this is of interest to you), which show HypoBoostrap's robustness across datasets and models.
> - **It is common that the models performing better on any reasoning tasks typically consume much more tokens.** Similar conclusions can be confirmed in the literatures on Math reasoning, code generation, etc.
> - **Our framework has decreasing marginal cost in terms of refinement iterations**, since only $H_F$ is regenerated in each refinement loop and the agents related to $H_O, H_R$ only use once at the very beginning. Note that HR consumes more tokens than us on MiniARC for $T=3$.
>
>
>
>
>
> > **Question 2**: How robust is the framework to flawed or uninformative intermediate hypotheses (HO,HR)? Since the refinement loop only targets HF​, are there cases where feedback should instead revise the more foundational HO or HR ?
>
> Our framework may indeed produce flawed or uninformative intermediate HO \& HR due to LLMs' hallucination, but $\texttt{ObjectHypothesisInconsistencyEliminator}$ and $\texttt{RelationalHypothesisInconsistencyEliminator}$ can eliminate several flawed HO \& HR (see Figure 3 for examples).
> We choose to eliminating them, instead of refining them, to reduce costs.
> However, the remaining flawed and uninformative HO \& HR indeed may still exist.
>
> To further improve these *bad* HO \& HR, we may have two choices, each of which has drawbacks.
> 1. If refining them in the current refinement loop, HO \& HR has no explicit optimizing signal since the feedback (passing the unit test or not) does not directly tell which HO \& HR are *bad* and how bad they are.
> 2. If refining them by simply ask LLMs to refine them, we also don't know how to tell which are *good* HO \& HR.
>
> Hence, it may be an interesting future work to enhance these intermediate hypotheses.
>
>
>
>
>
> > **Weakness 4**: The empirical results are reported without error bars or statistical significance tests. The authors note this is due to the high cost of replicating experiments on GPT-4.
>
> We apologize for the oversight in omitting the reporting of the decoding temperature of the LLMs.
> In our framework, all the LLMs using temperature 0 for generating texts.
> Therefore, **there is no randomness in our framework**.
> Error bars or statistical significance are thus unnecessary.
>
> Our small-scale preliminary experiments also show that our framework is stable across repeated experiments, and that baselines have little variance in experimental results.
>
>
>
>
>
> > **Weakness 6**: There is a minor inaccuracy in the submitted checklist regarding the declaration of LLM usage , which should be corrected to reflect that LLMs are a core component of the methodology.
>
> Thanks for pointing out our negligence in the checklist. We will correct it according to your suggestion.
>
>
>
> > **Limitation 2**: Clarify the framework’s scope and generality. HypoBootstrap rests on converting hypotheses into executable unit tests. The authors should explicitly state this dependency and discuss its applicability (or breakdown) when such tests cannot be constructed—for example, in open-ended text-only inductive tasks—so readers understand where the method may fail or need adaptation.
>
> Thanks for pointing out this important limitation.
> Our framework (and all the baselines) leverages unit tests, or more generally, a way to verify the correctness of the proposed hypotheses.
> In some real-world scenarios of inductive reasoning, such as scientific research (which is one of the largest applications of inductive reasoning), verifying the correctness of hypotheses can be costly due to the expensive experimental instruments.
> And, as you mentioned, in open-ended text-only inductive tasks, unit tests may be impossible.
>
> We will thoroughly discuss this limitation in the revised paper.
>
>
>
>
>
>
>
> Thank you again for the thorough review. We would be glad to provide any additional information if needed.

---

> > ### Author Response · Authors · 2025-08-01
> > **Minor correction of Rebuttal**
> >
> > For the tables reporting the number of tokens consumed by each method, the units are per thousand tokens
> >
> > In addition, we mistakenly added the results of MoC into the responses for you, which is actually requested by other reviewers. We apologize for this mistake.
> >
> > See our response to Reviewer tLyE for details of MoC's experiments if this is of interest to you.

---

> > > ### Comment · Area_Chair_aq4y · 2025-08-05
> > >
> > > The authors have provided a rebuttal to your comments, and it's an important part of the review process to give their response careful consideration. Please take a moment to review their rebuttal and provide any follow-up comments. This will help ensure there’s sufficient time for discussion and any necessary follow-up.
> > >
> > > Best regards,
> > >
> > > AC

---

> > ### Comment · Reviewer_6r5u · 2025-08-06
> >
> > Thank you for your detailed rebuttal. With my main concerns addressed, I’ll maintain my positive evaluation.

---

> > > ### Author Response · Authors · 2025-08-07
> > >
> > > Thank you for maintaining the positive evaluation of our work, and we appreciate again for all your efforts in reviewing our manuscript.

---

### Official Review · Reviewer_38v8 · 2025-07-02

**Clarity:** 2
**Significance:** 3
**Originality:** 3
**Rating:** 4
**Confidence:** 3

**Summary:**

The paper provides a formal logic description of a pipeline of LLMs for inductive reasoning - finding the underlying patterns given examples.
The LLMs first generate hypotheses, then verify them themselves.
Hypotheses are first generated for each input object, then for each input pair, finally for all the pairs together.
Evaluation is performed on 4 inductive reasoning datasets.

**Questions:**

What do you mean by "We use the 100 tasks filtered by HR"?

**Ethical Concerns:**

["NO or VERY MINOR ethics concerns only"]

**Final Justification:**

Thank you for the response. It's great that you provided the additional results with MoC. I think it's important to also highlight the differences and contributions in the main text. The clarity aspect is more difficult to assess based on the response, a new version of the paper would be needed. My score already leans positive and I will keep it the same.

**Limitations:**

Limitations are not discussed. The paper only mentions that the proposed method "cannot completely solve LLM’s hallucination problem".

**Paper Formatting Concerns:**

-

**Quality:**

3

**Strengths And Weaknesses:**

The pipeline of iterative hypothesis generation and self-checking seems to be a good idea.
Results show improvements over the chosen baseline.

The paper presentation is quite difficult to follow.
Everything is presented in an overly complex formal logic representation and notation, while the system is just a sequence of prompted LLMs and there is no way to guarantee that the LLM outputs actually follow that formal logic.
The structure of the paper could also be improved.

While MoC is claimed to be concurrent work, it has been on arxiv since December 2024. NeurIPS guidelines specify that appearing on arxiv more than 2 months earlier does not count as concurrect work. Therefore a more thorough comparison should be performed to that work.

---

> ### Author Rebuttal · Authors · 2025-07-31
>
> # Reviewer 38v8
>
> We sincerely thank you for the insightful comments and constructive feedback, which are highly valuable in improving our manuscript.
> We also appreciate your recognition of our idea and of the experimental improvement of our method.
>
> In the following, we address each of your questions in detail, point by point.
>
>
>
>
> > **Question**: What do you mean by "We use the 100 tasks filtered by HR"?
>
> To conduct fair comparison, our experimental setup, including datasets, exactly follows Hypothesis Refinement (HR) [19].
> In the Appendix A of HR, HR's author mentioned that they filtered a evaluation subset from the original SCAN dataset [11] via some heuristic methods.
> The filtered evaluation subset has 100 SCAN tasks.
>
> We will further clarify this in the revised paper.
>
>
>
> > The paper presentation is quite difficult to follow. Everything is presented in an overly complex formal logic representation and notation, while the system is just a sequence of prompted LLMs and there is no way to guarantee that the LLM outputs actually follow that formal logic. The structure of the paper could also be improved.
>
> Thanks for pointing out the clarity issue and the important limitation that LLMs cannot strictly adhere to formal logic.
> - For the clarity issue, we will try our best to revised the structure of the paper and make the description of our method easier to follow.
> - For the limitation, we acknowledge that LLMs cannot strictly adhere to formal logic, and that is one of the major reasons why our method cannot achieve 100% accuracy. In Section 6 and Appendix F, we already acknowledge that our method cannot completely tackles LLM's hallucination in inductive reasoning. We will further emphasize the limitation you pointed out in the revised paper.
>
> In addition, although formalizing our method in formal logic increases difficulty in understanding, it has several advantages:
>
> 1. Our proposed hypothesis confirmation method comes from a theoretical principle from philosophy of science. i.e., Glymour's theory of bootstrap confirmation, **which is originally rigorously discussed in formal logic**. This principle confirmation method makes our method theoretically-grounded and not simply heuristic.
> Moreover, our proposed hypothesis generation method comes from the observation of the difference between natural language and formal language.
> Hence, formalizing our method in formal logic makes sense.
> 2. **Formalizing inductive reasoning in formal logic is common in AI** (i.e., Inductive Logic Programming; see Line 29-35) **and philosophy of science**. It is easier for researchers from other domains related to inductive reasoning to understand our method.
>
> We will try our best to make our paper easier to follow for broader audience.
>
>
>
> > While MoC is claimed to be concurrent work, it has been on arxiv since December 2024. NeurIPS guidelines specify that appearing on arxiv more than 2 months earlier does not count as concurrect work. Therefore a more thorough comparison should be performed to that work.
>
> Thanks for pointing out our misunderstanding of the definition of concurrent work.
> We add experiments on MoC.
>
>
>
> [GPT-4; $T=1$]
> | Method | ACRE (Raw\Task) | List Fns (Raw\Task) | MiniSCAN (Raw\Task) | MiniARC (Raw\Task) | Avg. (Raw\Task) |
> | - | - | - | - | - | - |
> | HR | 78.2\45.0 | 51.6\42.4 | 77.0\46.0 | 5.9\3.8 | 53.2\34.3 |
> | MoC | 74.0\34.0 | 53.0\43.2 | 60.4\21.0 | 6.7\4.6 | 48.5(-4.7)\25.7(-8.6) |
> | HB (ours) | **84.8**\\**57.0** | **57.9**\\**47.6** | **94.9**\\**82.0** | **11.0**\\**6.9** | **62.1(+8.9)**\\**48.4(+14.1)** |
>
>
>
> [GPT-4; $T=3$]
> | Method | ACRE (Raw\Task) | List Fns (Raw\Task) | MiniSCAN (Raw\Task) | MiniARC (Raw\Task) | Avg. (Raw\Task) |
> | - | - | - | - | - | - |
> | HR | 77.8\47.0 | 61.7\52.8 | **98.2**\\**95.0** | 10.1\6.9 | 62.0\50.4 |
> | MoC | 73.0\39.0 | 61.9\50.8 | 87.3\64.0 | 10.8\\**8.5** | 58.3(-3.7)\40.6(-9.8) |
> | HB* (ours) | **79.0**\\**55.0** | **64.3**\\**53.6** | **98.2**\89.0 | **12.1**\\**8.5** | **63.4(+1.4)\51.5(+1.1)** |
>
>
> It is interesting to see the unstable performance of MoC.
> Since MoC did not release its code, we implement it according to its paper published in NAACL 2025.
> On ListFns and MiniARC, which are also used in MoC's paper, MoC outperforms HR or has comparable performance, which is inline with the results in their paper.
> However, On ACRE and MiniSCAN, which are not used in MoC's paper, MoC show very unstable performance across tasks and models. Sometimes MoC is even significantly worse than HR.
> - ACRE and MiniSCAN all require generating hypotheses under a specific format and the correct hypotheses are usually highly composite, i.e., the output (in evidence) must be obtained by applying the rules in combination. This significantly differs from ListFns and MiniARC, which makes ACRE and MiniSCAN a better benckmark.
> - **The comparison between the robust performance of our framework and the unstable performance of MoC reveals the benefit of our framework.**
>
>
>
> See our response to Reviewer tLyE for results on DeepSeek-V3, If this is of interest to you.
> MoC outperforms HR but underperforms our framework on DeepSeek-V3.
>
>
>
>
> > **Limitations**: Limitations are not discussed. The paper only mentions that the proposed method "cannot completely solve LLM’s hallucination problem".
>
> Thanks for pointing out this limitation.
> Here are limitations summarized from our new observation and other reviewers.
> 1. Our framework (and all baselines) requires a way to verify generated hypotheses, e.g., unit tests.
> However, unit test is not always available in real-world scenario, such as open-ended text-based induction.
> In scientific research, which is one of the largest applications of inductive reasoning, hypothesis verification is costly due to expensive experimental instruments.
> 2. Our framework consumes more tokens than baselines.
>     - However, the robust performance across tasks and models deserves more tokens. (see our response to Reviewer tLyE for detailed discussion if this is of interest to you)
>
> We will emphasize these limitations in the revised paper.
>
>
>
>
> Thank you again for the thorough review. We would be glad to provide any additional information if needed.

---

> ### Author Response · Authors · 2025-08-01
> **Minor correction of Rebuttal**
>
> For the experiments on MoC, there is an unclearness in the tables' header.
>
> $T=1$ or $T=3$ indicates HR and HB (or HB*) refine the (functional) hypothesis by 1 or 3 iterations. Each method generates only one hypothesis per iteration.
>
> For MoC, there is no such hyperparameter since there is no refinement. A relevant hyperparameter is $K$, indicating the number of hints generated before generating hypotheses, which is equal to the number of generated hypotheses.
>
> Hence, to make fair comparison, we set $K=1$ for $T=1$ and $K=3$ for $T=3$.
>
> **This is to ensure all methods generate the same number of (functional) hypotheses and leverage the same times of unit tests.**
> (All methods leverage unit test once generate a (functional) hypothesis.)

---

> > ### Comment · Area_Chair_aq4y · 2025-08-05
> >
> > The authors have provided a rebuttal to your comments, and it's an important part of the review process to give their response careful consideration. Please take a moment to review their rebuttal and provide any follow-up comments. This will help ensure there’s sufficient time for discussion and any necessary follow-up.
> >
> > Best regards,
> >
> > AC

---

> ### Author Response · Authors · 2025-08-07
>
> Dear Reviewer,
>
> Thank you again for your insightful and valuable comments on our manuscript. We have carefully responded to your questions and added new experiments. We hope these revisions fully resolve your concerns.
>
> If you find that the additional results and analysis satisfactorily answer your questions, we would be grateful if you could revisit your rating to reflect your latest evaluation.
> If any new clarification is needed, we remain at your disposal to provide all of them.
>
> As the discussion deadline is approaching, your timely feedback would be greatly appreciated and will help ensure that all remaining questions are thoroughly addressed. We apologize for contacting you directly, and we appreciate again for all your efforts devoted to our work.
>
> Best regards,
>
> All authors

---

### Official Review · Reviewer_tLyE · 2025-07-03

**Clarity:** 2
**Significance:** 2
**Originality:** 2
**Rating:** 4
**Confidence:** 3

**Summary:**

The paper focuses on the task of inductive reasoning, where a model needs to infer a pattern (usually a mapping) from a given set of observations (typically multiple input-output pairs). Inspired by a hypothesis confirmation theory from the philosophy of science, the authors propose an LLM-based framework in which hypotheses of varying granularity are proposed and verified progressively. Specifically, in the first stage, the LLM is asked to describe each input or output individually. Then, it is asked to identify the patterns between a single input-output pair. Finally, the pattern governing all input-output pairs is inferred based on the information from the previous stages. Experiments have shown that the method is effective across various inductive reasoning benchmarks.

**Questions:**

1. It is important to report the average computation cost for each problem across all datasets considered.
2. Comparison with MoC would be helpful to show the effectiveness of the method.
3. It is also valuable to test the proposed method on more LLMs to see the robustness of the method.

**Ethical Concerns:**

["NO or VERY MINOR ethics concerns only"]

**Final Justification:**

The authors propose an effective method for inductive reasoning using large language models, drawing inspiration from the hypothesis confirmation theory. Extensive experiments have demonstrated the effectiveness of the method on a wide range of inductive reasoning tasks using different large language models.

**Limitations:**

Yes.

**Paper Formatting Concerns:**

No.

**Quality:**

2

**Strengths And Weaknesses:**

# Strengths
- The paper is mostly clearly written.
- Leveraging ideas from hypothesis confirmation theory is interesting.

# Weaknesses
1. It is uncertain whether the comparison with the baseline is fair. It is unclear how many hypotheses the proposed method (HB) and the baseline method (HR) generate. Since the two methods differ significantly in how hypotheses are generated, it is essential to report the computational cost of each method for a given problem. This is a good measurement of the compute needed to solve each problem, considering the context length and the number of generated tokens. But unfortunately, this information is missing in the paper.
2. The method shows only marginal improvement (1.1% on average) over the simple baseline HR when the number of iterations increases to 3. This suggests that the effectiveness of the proposed algorithm is limited. Given that the method is relatively complex, this also raises questions about its generalizability to other domains or its performance when provided with more compute.
3. Although MoC was published one month before the NeurIPS submission deadline, its arXiv version appeared in December 2024. According to the policy stated here, this does not qualify as concurrent work. While I believe the methods are significantly different, MoC would still serve as a simpler and strong baseline for comparison.

## Minor Points
- The experiments section is poorly written:

  - The meaning of T is not defined. It likely refers to the number of refinement rounds, as in previous work (HR), but this is not explained.

  - The number of hypotheses generated for the HR baseline is not reported.

---

> ### Author Rebuttal · Authors · 2025-07-31
>
> We sincerely thank you for the insightful comments and constructive feedback, which are highly valuable in improving our manuscript.
> We also appreciate your recognition of the integration of hypothesis confirmation theory and of the demonstrated effectiveness of our method.
>
> In the following, we address each of your questions in detail, point by point.
>
>
>
>
> > **Weakness Minor Points**:
> > The experiments section is poorly written:
> > - The meaning of T is not defined. It likely refers to the number of refinement rounds, as in previous work (HR), but this is not explained.
> > - The number of hypotheses generated for the HR baseline is not reported.
>
> We apologize for the unclearness. Here are clarifications.
> - $T$ refers to the number of refinement loop, which is defined in Section 4.3. We will emphasize the meaning of $T$ in experiments section.
> - The number of hypotheses generated for the HR is 1 per refinement iteration. In Section 5.1, we claim that *we only allow one functional hypothesis to be generated per iteration*. In the revised paper, we will clarify that this is also valid for baselines.
>
>
>
>
> > **Question 2**: Comparison with MoC would be helpful to show the effectiveness of the method.
> >
> > **Weakness 3**: Although MoC was published one month before the NeurIPS submission deadline, its arXiv version appeared in December 2024. According to the policy stated here, this does not qualify as concurrent work. While I believe the methods are significantly different, MoC would still serve as a simpler and strong baseline for comparison.
> >
> > **Question 3**: It is also valuable to test the proposed method on more LLMs to see the robustness of the method.
>
> Thanks for pointing out our misunderstanding of the definition of concurrent work.
> We add experiments on MoC.
> In addition, also according to your suggestion, we also add experiments on DeepSeek-V3 (671B-A37B). Due to the limited time of rebuttal period, we are still running experiments on other LLMs, but the results on DeepSeek-V3 is already convincing to show the robustness of our method.
>
> The results are shown below.
> To conduct fair comparison, we let all methods generate the same number of functional hypotheses (i.e., set $K=3$ for MoC).
> This number is linearly correlated with the number of required unit tests in all methods, which is an important limitation in some real-world applications (see our response to your Weakness 2).
>
>
>
>
> [GPT-4; 1 hypothesis]
> | Method | ACRE (Raw\Task) | List Fns (Raw\Task) | MiniSCAN (Raw\Task) | MiniARC (Raw\Task) | Avg. (Raw\Task) |
> | - | - | - | - | - | - |
> | HR | 78.2\45.0 | 51.6\42.4 | 77.0\46.0 | 5.9\3.8 | 53.2\34.3 |
> | MoC | 74.0\34.0 | 53.0\43.2 | 60.4\21.0 | 6.7\4.6 | 48.5(-4.7)\25.7(-8.6) |
> | HB (ours) | **84.8**\\**57.0** | **57.9**\\**47.6** | **94.9**\\**82.0** | **11.0**\\**6.9** | **62.1(+8.9)**\\**48.4(+14.1)** |
>
>
>
> [GPT-4; 3 hypotheses]
> | Method | ACRE (Raw\Task) | List Fns (Raw\Task) | MiniSCAN (Raw\Task) | MiniARC (Raw\Task) | Avg. (Raw\Task) |
> | - | - | - | - | - | - |
> | HR | 77.8\47.0 | 61.7\52.8 | **98.2**\\**95.0** | 10.1\6.9 | 62.0\50.4 |
> | MoC | 73.0\39.0 | 61.9\50.8 | 87.3\64.0 | 10.8\\**8.5** | 58.3(-3.7)\40.6(-9.8) |
> | HB* (ours) | **79.0**\\**55.0** | **64.3**\\**53.6** | **98.2**\89.0 | **12.1**\\**8.5** | **63.4(+1.4)\51.5(+1.1)** |
>
>
>
>
> [DeepSeek-V3; 1 hypothesis]
> | Method | ACRE (Raw\Task) | List Fns (Raw\Task) | MiniSCAN (Raw\Task) | MiniARC (Raw\Task) | Avg. (Raw\Task) |
> | - | - | - | - | - | - |
> | HR | 72.0\37.0 | 46.4\36.4 | 25.8\21.0 | 9.5\7.7 | 38.4\25.5 |
> | MoC | 70.5\34.0 | 52.5\40.8 | 30.2\11.0 | 9.2\6.2 | 40.6(+2.2)\23.0(-2.5) |
> | HB (ours) | **74.5**\\**47.0** | **55.6**\\**42.0** | **69.8**\\**47.0** | **16.7**\\**10** | **54.2(+15.8)\36.5(+11.0)** |
>
>
>
> [DeepSeek-V3; 3 hypotheses]
> | Method | ACRE (Raw\Task) | List Fns (Raw\Task) | MiniSCAN (Raw\Task) | MiniARC (Raw\Task) | Avg. (Raw\Task) |
> | - | - | - | - | - | - |
> | HR | 77.3\\**53.0** | 52.9\43.6 | 34.3\24.0 | 11.4\9.2 | 44.0\32.5 |
> | MoC | 73.5\43.0 | 59.2\48.4 | 71.9\51.0 | 14.9\11.5 | 54.9(+10.9)\38.5(+6.0) |
> | HB* (ours) | **77.5**\\**53.0** | **62.8**\\**48.8** | **74.0**\\**53.0** | **21.8**\\**14.6** | **59.0(+15.0)\42.4(+9.9)** |
>
>
> We can draw the following additional conclusions:
> - **Our framework significantly outperforms HR and MoC**: Compared to HR, HB achieves an average improvement in task accuracy across four datasets of: +14.1 (GPT-4; 1 hypo), +11.0 (DeepSeek-V3; 1 hypo), +9.9 (DeepSeek-V3; 3 hypo).
> Compared to MoC, HB's improvement is much higher on GPT-4 and [DeepSeek-V3; 1 hypo]. For [DeepSeek-V3; 3 hypos], HB also outperforms MoC about 4 percentage points.
> - **Our framework is robust across datasets and models**: Among all the results above, HB achieves the best performance across all datasets and models for both raw accuracy and task accuracy, except for GPT-4's task accuracy on MiniSCAN.
> - **MoC is unstable across tasks and models**: It is interesting to see the unstable performance of MoC. Since MoC did not release its code, we implement it according to its paper published in NAACL 2025. On ListFns and MiniARC, which are also used in MoC's paper, MoC outperforms HR or achieves comparable performance, which is inline with the results in their paper. However, On ACRE and MiniSCAN, which are not used in MoC's paper, MoC shows very unstable performance across tasks and models. Sometimes MoC is even significantly worse than HR.
>     - ACRE and MiniSCAN all require generating hypotheses under a specific format and the correct hypotheses are usually highly composite, i.e., the output (in evidence) must be obtained by applying the rules in combination. This significantly differs from ListFns and MiniARC, which makes ACRE and MiniSCAN a better benckmark.
>     - **The comparison between the robust performance of our framework and the unstable performance of MoC reveals the benefit of our framework.**
>
>
>
>
>
>
>
>
>
>
>
> > **Question 1**: It is important to report the average computation cost for each problem across all datasets considered.
> >
> > **Weakness 1**: It is uncertain whether the comparison with the baseline is fair. It is unclear how many hypotheses the proposed method (HB) and the baseline method (HR) generate. Since the two methods differ significantly in how hypotheses are generated, it is essential to report the computational cost of each method for a given problem. This is a good measurement of the compute needed to solve each problem, considering the context length and the number of generated tokens. But unfortunately, this information is missing in the paper.
>
> Here is the token consumed by all methods.
> (It's slightly simpler to count tokens on DeepSeek-V3, considering the limited rebuttal period.)
>
>
>
> [DeepSeek-V3; 1 hypothesis] (P for prompt tokens; C for completion tokens)
> | Method | ACRE (P\C) | List Fns (P\C) | MiniSCAN (P\C) | MiniARC (P\C) |
> | - | - | - | - | - |
> | HR | 22\6 | 103\120 | 43\29 | 99\215 |
> | MoC | 46\24 | 209\67 | 80\72 | 152\50 |
> | HB (ours) | 400\69 | 2409\633 | 1201\245 | 991\283 |
>
>
>
> [DeepSeek-V3; 3 hypotheses]
> | Method | ACRE (P\C) | List Fns (P\C) | MiniSCAN (P\C) | MiniARC (P\C) |
> | - | - | - | - | - |
> | HR | 56\10 | 477\352 | 331\77 | 565\532 |
> | MoC | 98\23 | 439\81 | 177\142 | 317\90 |
> | HB* (ours) | 572\82 | 3121\745 | 1642\305 | 1616\407 |
>
>
>
> Our framework does consume about 1~10 times tokens than baseline methods.
> However, this is not surprising since
> - **Our framework is much more robust across models and datasets than baselines**, as demonstrated above.
> - **It is common that the models performing better on any reasoning tasks typically consume much more tokens.** Similar conclusions can be confirmed in the literatures on Math reasoning, code generation, etc.
> - **Our framework has decreasing marginal cost in terms of refinement iterations**, since only $H_F$ is regenerated in each refinement loop and the agents related to $H_O, H_R$ only use once at the very beginning.
>
>
>
>
> In addition, **the experimental comparison is fair since the number of functional hypotheses generated by HR, MoC, and our framework are the same**, as stated above.
> An important reason for ensuring same number of functional hypotheses is that, for all methods reported, this number is linearly correlated with the number of required unit tests in all methods, which is an important limitation in some real-world applications (see our response to your Weakness 2).
>
>
>
>
>
>
>
>
>
> > **Weakness 2**: The method shows only marginal improvement (1.1% on average) over the simple baseline HR when the number of iterations increases to 3. This suggests that the effectiveness of the proposed algorithm is limited. Given that the method is relatively complex, this also raises questions about its generalizability to other domains or its performance when provided with more compute.
>
> According to our responses above, we can draw the following conclusions.
> - **Our framework shows significant performance gain on DeepSeek-V3 for $T=3$**, revealing that the relative small gain on GPT-4 may be model-related.
> - **Performance on $T=1$ is more important in the sense that no hypothesis verification is required.** As pointed out by Reviewer 6r5u, the unit test, which is used as hypothesis verification in all methods reported in the paper, is not always available in real-world applications. In scientific research, which is one of the largest applications of inductive reasoning, hypothesis verification is also costly due to expensive experimental instruments. Hence, generating high-quality hypothesis without verification is a critical ability. We apologize for not emphasizing this importance in the paper and will revised the paper accordingly.
>
>
>
>
>
>
>
>
>
>
>
> Thank you again for the thorough review. We would be glad to provide any additional information if needed.

---

> > ### Comment · Reviewer_tLyE · 2025-08-06
> >
> > Thanks to the authors for conducting extensive experiments and addressing my concerns. While I believe using 10x more tokens is a crucial limitation of the proposed method, the proposed method and the extensive evaluations could still benefit the community. Thus, I have raised my score to Weak Accept. I strongly encourage the authors to incorporate all results in the rebuttal into the revision, especially the token count analysis.

---

> > > ### Author Response · Authors · 2025-08-06
> > >
> > > Thanks for raising the score. We will include the additional experimental results in the revised paper, and emphasize the limitation of more token consumption. We appreciate again for all your efforts in reviewing our paper.

---

> > > ### Author Response · Authors · 2025-08-07
> > >
> > > Dear Reviewer,
> > >
> > > Thank you again for raising the rating to Weak Accept.
> > >
> > > However, in our reviewing system, your rating is still displayed as Borderline Reject.
> > > In addition, according to Program Chairs' email, the rating should disappear for authors if reviewer has updated the rating, but we can still see your rating (which is Borderline Reject).
> > >
> > > Hence, it seems that there may be a bug in the reviewing system.
> > > **We would be grateful if you could help ensure the rating has already been correctly updated.**
> > >
> > > Best regards,
> > >
> > > All authors

---

> ### Author Response · Authors · 2025-08-01
> **Minor correction of Rebuttal**
>
> # Correction 1
>
> For the experiments on MoC, the hyperparameter $K$ is set to 1 if HR and HB set $T=1$ (where we have marked *1 hypothesis* in Table's header). The hyperparameter $K$ is set to 3 if HR and HB set $T=3$ (where we have marked *3 hypotheses* in Table's header).
>
> **This is to ensure all methods generate the same number of (functional) hypotheses and leverage the same times of unit tests.**
>
> Detailed explanation:
> - $T=1$ or $T=3$ indicates HR and HB (or HB*) refine the (functional) hypothesis by 1 or 3 iterations. Each method generates only one hypothesis per iteration.
> - For MoC, there is no such hyperparameter since there is no refinement. A relevant hyperparameter is $K$, indicating the number of hints generated before generating hypotheses, which is equal to the number of generated hypotheses.
>
>
>
>
> # Correction 2
>
> For the tables reported the number of tokens consumed by each methods, the units are per thousand tokens.

---

> > ### Comment · Area_Chair_aq4y · 2025-08-05
> >
> > The authors have provided a rebuttal to your comments, and it's an important part of the review process to give their response careful consideration. Please take a moment to review their rebuttal and provide any follow-up comments. This will help ensure there’s sufficient time for discussion and any necessary follow-up.
> >
> > Best regards,
> >
> > AC

---

### Official Review · Reviewer_vRQJ · 2025-07-06

**Clarity:** 3
**Significance:** 3
**Originality:** 3
**Rating:** 4
**Confidence:** 3

**Summary:**

The paper proposed a framework addressing two core steps in inductive reasoning: hypothesis generation and hypothesis confirmation. For hypothesis generation, it incrementally constructs object-level, relational, and then functional hypotheses in a bottom-up manner, helping LLMs move from simple patterns to more complex generalizations. For hypothesis confirmation, it employs Glymour’s bootstrap confirmation theory, a philosophy-of-science-based method that ensures sets of hypotheses are jointly consistent, supported by evidence, and non-trivial. Experiments across diverse reasoning tasks show that HypoBootstrap significantly reduces hallucinations and improves accuracy over prior LLM-based methods.

**Questions:**

The main points are covered in the strength and weakness. Some actionable items could be
1. Experiment with some open source models and reveal if the solution is generalizable. If the open-source models have gap, it would be interesting to see some analysis on the gap.
2. Some error case study would be interesting, especially the hallucination issues before and after the solution is applied.

**Ethical Concerns:**

["NO or VERY MINOR ethics concerns only"]

**Limitations:**

Yes.

**Quality:**

3

**Strengths And Weaknesses:**

Strength 1: Motivation is clear. The paper starts by clearly identifying a key challenge: LLMs are capable of generating hypotheses in natural language but tend to hallucinate and lack structured guidance during inductive reasoning. Though I have some hypothesis that the inductive reasoning in natural language might already be sufficient in the era of LLM.

Strength 2: The paper uses Glymour’s theory of bootstrap instead of relying on ad hoc metrics, which is unique and principled. The authors adopt a confirmation method rooted in the philosophy of science, ensuring hypotheses are consistent, non-trivial, and jointly supported by evidence.
Strength 3: The framework is evaluated on four types of inductive reasoning tasks—causal induction, concept learning, grammar learning, and abstract reasoning. HypoBootstrap outperforms strong baselines with a clear performance gain. Code repo is available to publish, which confirms the reproducibility of the paper.

The weakness of the paper:
W1: The experiment mainly adopts GPT4, instead of open source models. What are the limitations of open source models? It would be interesting to see such study.
W2: While ablations are provided for object/relational hypotheses, there is no detailed ablation isolating the impact of Glymour’s bootstrap confirmation. It’s hard to tell how much of the gain comes from confirmation vs generation improvements.
W3. Some study in failure cases or hallucination types is missing. Though the paper aims to reduce hallucination, it does not provide a systematic analysis of when and how hallucinations still occur, nor categorize failure modes.

---

> ### Author Rebuttal · Authors · 2025-07-31
>
> # Reviewer vRQJ
>
> We sincerely thank you for the insightful comments and constructive feedback, which are highly valuable in improving our manuscript.
> We also appreciate your recognition of the principled foundation, uniqueness, and high effectiveness of our method.
>
> In the following, we address each of your questions in detail, point by point.
>
>
> > **Question 1**: Experiment with some open source models and reveal if the solution is generalizable. If the open-source models have gap, it would be interesting to see some analysis on the gap.
> >
> > **Weakness 1**: The experiment mainly adopts GPT4, instead of open source models. What are the limitations of open source models? It would be interesting to see such study.
>
>
> We add experiments on open-source DeepSeek-V3 (671B-A37B).
> Due to the limited time of rebuttal period, we are still running experiments on other LLMs, but the results on DeepSeek-V3 is already convincing to show the generalization ability of our method.
>
>
>
>
> [$T=1$]
> | Method | ACRE (Raw\Task) | List Fns (Raw\Task) | MiniSCAN (Raw\Task) | MiniARC (Raw\Task) | Avg. (Raw\Task) |
> | - | - | - | - | - | - |
> | HR | 72.0\37.0 | 46.4\36.4 | 25.8\21.0 | 9.5\7.7 | 38.4\25.5 |
> | HB (ours) | **74.5**\\**47.0** | **55.6**\\**42.0** | **69.8**\\**47.0** | **16.7**\\**10** | **54.2(+15.8)\36.5(+11.0)** |
>
>
>
> [$T=3$]
> | Method | ACRE (Raw\Task) | List Fns (Raw\Task) | MiniSCAN (Raw\Task) | MiniARC (Raw\Task) | Avg. (Raw\Task) |
> | - | - | - | - | - | - |
> | HR | 77.3\\**53.0** | 52.9\43.6 | 34.3\24.0 | 11.4\9.2 | 44.0\32.5 |
> | HB* (ours) | **77.5**\\**53.0** | **62.8**\\**48.8** | **74.0**\\**53.0** | **21.8**\\**14.6** | **59.0(+15.0)\42.4(+9.9)** |
>
>
> These results not only show the significant performance gain for $T=1$ (the same for GPT-4), but also show significant performance gain for $T=3$.
> The robustness and generalization ability across models and datasets are well proven.
>
>
>
>
>
> See our response to Reviewer tLyE for results of MoC (if it is interesting for you), which we misunderstood as concurrent work but actually not.
>
>
>
>
>
>
>
> > **Question 2**: Some error case study would be interesting, especially the hallucination issues before and after the solution is applied.
> >
> > **Weakness 3**: Some study in failure cases or hallucination types is missing. Though the paper aims to reduce hallucination, it does not provide a systematic analysis of when and how hallucinations still occur, nor categorize failure modes.
>
> Due to the 'black box' nature of current LLMs and the complexity of natural language, it is difficult to identify the patterns and mechanisms behind hallucination. We have uncovered some potential patterns by analyzing the categories of error cases.
>
> We examined the failure cases and categorized them into distinct failure modes.
> Note: The citation number is the same as in the paper.
>
> **1.ACRE**
> The reasoning query modes in the ACRE dataset can be categorized into four types[26], including: direct, indirect, screening-off, and backward-blocking. We found several failure cases in backward-blocking reasoning  (i.e., in addition to being a Blicket or non-Blicket, an object’s Blicketness could also be undetermined, which occurs when the object is not directly tested, but can activate the machine together with other potential Blickets).
> An example is given below:
>
> {"idx": 14, train":[{"input": ["blue rubber cube"], "output": "off"}, {"input": ["red rubber cube"], "output": "on"}, {"input": ["red rubber cube", "blue rubber cube"], "output": "on"}, {"input": ["red metal cube", "cyan rubber cube"], "output": "off"}, {"input": ["gray metal cube", "blue rubber sphere", "red metal cube"], "output": "on"}, {"input": ["yellow rubber cylinder"], "output": "off"}],
> "test": [{"input": ["cyan rubber cube"], "output": "off", "type": "direct"}, {"input": ["gray metal cube"], "output": "undetermined", "type": "potential"}, {"input": ["blue rubber cube", "blue rubber sphere"], "output": "undetermined", "type": "potential"}, {"input": ["blue rubber cube", "gray metal cube"], "output": "undetermined", "type": "potential"}]}
>
> Generated Rule: {"blue rubber sphere": "off"}
> Correct Rule: {"blue rubber sphere": "undetermined"}
>
> Hallucinations occur when the model analyzes the fifth task:{"input": ["gray metal cube", "blue rubber sphere", "red metal cube"], "output": "on"}. While the 'blue rubber sphere' is not directly tested, it could potentially activate the machine together with other Blickets (like the 'gray metal cube'). Therefore, the Blicketness of the 'blue rubber sphere' should be 'undetermined'.
>
> **2.List Functions**
> List Functions tasks are organized into different groups [20], containing problems of varying difficulty to test for differences in:
> (1) recursive and non-recursive reasoning
> (2) the ability to handle exceptions
> (3) biases toward elements occurring early in a list.
> We found that hallucinations occur when dealing with biases toward elements occurring early in a list. An example is given below:
>
> {"idx": 84, "train": [{"input": "[2, 1, 40, 92, 26, 95, 45]", "output": "[40]"}, {"input": "[1, 8]", "output": "[8]"}, {"input": "[1, 4, 5, 80, 76, 0, 2, 71, 26]", "output": "[4]"}, {"input": "[1, 14, 3, 2, 78, 9]", "output": "[14]"}, {"input": "[6, 2, 9, 8, 44, 7, 3]", "output": "[3]"}, {"input": "[1, 53]", "output": "[53]"}, {"input": "[3, 6, 4, 93, 43]", "output": "[93]"}, {"input": "[3, 7, 55, 75, 5, 24, 14, 1]", "output": "[75]"}], "test": [{"input": "[9, 60, 11, 93, 4, 5, 8, 24, 1, 97]", "output": "[97]"}, {"input": "[2, 8, 47, 60, 0, 18, 5, 7, 3]", "output": "[47]"}, {"input": "[3, 72, 2, 75, 27, 6, 7, 4]", "output": "[75]"}, {"input": "[1, 62, 4]", "output": "[62]"}, {"input": "[5, 17, 1, 4, 3, 88, 68]", "output": "[88]"}, {"input": "[1, 56, 7, 30]", "output": "[56]"}, {"input": "[4, 6, 71, 0, 9, 86, 59, 29, 3]", "output": "[9]"}, {"input": "[5, 64, 3, 7, 65, 83, 54, 27, 4]", "output": "[83]"}]}
>
> Generated Rule: The output is the second highest number in the input list.
> Correct Rule: Remove all but element N + 1, N = element 1.
>
> **3.MiniSCAN**
> Tasks of MiniSCAN dataset contain two types of pseudowords, some pseudowords were primitive instructions that correspond to a single output symbol, while other pseudowords are interpreted as functions that need to be applied to arguments to construct the output [11].
> Functions can be categorized into three types:
> (1) Function 1 takes the preceding primitive as an argument and repeats its output three times; (2) Function 2 takes both the preceding primitive and following primitive as arguments, producing their outputs in a specific alternating sequence.
> (3) Function 3 takes both the preceding and following strings as input, processes them, and concatenates their outputs in reverse order.
> We found that hallucinations occur when the model extracts rules like Function 3 in cases where its arguments are generated by other functions. For example, the model generates the incorrect rule "##A spiirth ##B shruw ##C" → "##C ##A ##B ##C" when the correct rule should be "##A spiirth ##B shruw ##C" → "##C ##A ##B ##A."
>
> **4.MiniARC**
> As the most challenging dataset, MiniARC[1] aims to evaluate LLMs' human cognitive priors including:
> (1) Objectness priors (object cohesion, object persistence, and object influence via contact)
> (2) Goal-directedness prior
> (3) Numbers and Counting priors
> (4) Basic Geometry and Topology priors
> We found that our approach still underperforms on these tasks, as the inputs are limited to text-based matrices. In future work, we will explore whether incorporating visual information through multimodal LLMs could improve performance.
>
> > **Weakness 2**: While ablations are provided for object/relational hypotheses, there is no detailed ablation isolating the impact of Glymour’s bootstrap confirmation. It’s hard to tell how much of the gain comes from confirmation vs generation improvements.
>
> We also separately study the impact of the confirmation part and the generation part of HypoBootstrap in Section 5.2.
>
> - **Independent impact of the proposed generation in HypoBootstrap**:
> The variant $\widetilde{\textrm{HB}}$ removes all the inconsistency elimination and consistency validation from HypoBootstrap.
> The only remaining component of the confirmation part is the unit test ($\texttt{AuxiliaryConfirmer}$), which is also used in HR (the baseline).
>     - For $T=1$, the unit test isn't used at all; hence, the results of $\widetilde{\textrm{HB}}$ under $T=1$ in Table 1 shows the independent impact of the proposed generation method. Since $\widetilde{\textrm{HB}}$ outperforms HR significantly (+10.4 task accuracy), our proposed generation is very effective.
>     - For $T=3$, $\widetilde{\textrm{HB}}$ underperforms HR slightly, showing that the proposed generation is not good enough in multi-turn refinement setup.
>
> - **Impact of the proposed confirmation in HypoBootstrap**:
>     - From $T=1$, the results of HB (add back the inconsistency elimination and consistency validation) and $\widetilde{\textrm{HB}}$, show that the consistency related part of our proposed confirmation method is effective (+3.7 task accuracy).
>     - From $T=3$, comparing HR, $\widetilde{\textrm{HB}}$, and HB (or HB*), we found the confirmation part is indispensable for our proposed generation method.
>
> We will further revised the paper to clearly state these observations.
>
>
>
>
>
>
> Thank you again for the thorough review. We would be glad to provide any additional information if needed.

---

> > ### Comment · Area_Chair_aq4y · 2025-08-05
> >
> > The authors have provided a rebuttal to your comments, and it's an important part of the review process to give their response careful consideration. Please take a moment to review their rebuttal and provide any follow-up comments. This will help ensure there’s sufficient time for discussion and any necessary follow-up.
> >
> > Best regards,
> >
> > AC

---

> ### Author Response · Authors · 2025-08-07
>
> Dear Reviewer,
>
> Thank you again for your insightful and valuable comments on our manuscript. We have carefully responded to your questions and added new experiments. We hope these revisions fully resolve your concerns.
>
> If you find that the additional results and analysis satisfactorily answer your questions, we would be grateful if you could revisit your rating to reflect your latest evaluation.
> If any new clarification is needed, we remain at your disposal to provide all of them.
>
> As the discussion deadline is approaching, your timely feedback would be greatly appreciated and will help ensure that all remaining questions are thoroughly addressed. We apologize for contacting you directly, and we appreciate again for all your efforts devoted to our work.
>
> Best regards,
>
> All authors

---

### Decision · Program_Chairs · 2025-09-17

**Decision:**

Accept (poster)

**Comment:**

The paper presents a novel approach that adapts Glymour's bootstrap confirmation theory to guide inductive reasoning in LLMs. All reviewers are positive with the originality, technically robustness, rigorous experimental validation, and clear presentation.